# FACT depletion demonstrates a role for nucleosome organization in TAD formation

Clemens Mauksch [1,2], Yi Zhu [1,2], Taras Velychko [3], Spyridoula Sagropoulou [1,2], Abrar Aljahani[1,2,4], Shyam Ramasamy [1,2], Kristina Žumer [3] & A Marieke Oudelaar [1✉]

## Abstract

Mammalian genomes are organized into distinct chromatin structures, which include small-scale nucleosome arrays and large-scale topologically associating domains (TADs). The mechanistic interplay between chromatin structures across scales is poorly understood. Here, we investigate how changes in nucleosome organization impact TAD structure by studying the role of the histone chaperone facilitates chromatin transcription (FACT) in 3D genome organization. We show that FACT depletion perturbs TADs, causing decreased insulation and weaker CTCF loops. These changes in TAD structure cannot be attributed to changes in chromatin occupancy of CTCF or cohesin and occur specifically in transcribed regions of the genome, where we observe perturbed nucleosome organization in the absence of FACT. FACT depletion therefore allows us to separate the role of nucleosome organization and CTCF binding and to demonstrate that the organization of nucleosomes at TAD boundaries contributes to TAD formation.

**Keywords** 3D Genome Organization; Nucleosome; Topologically Associating Domain (TAD); FACT; CTCF
**Subject Category** Chromatin, Transcription & Genomics

## Introduction

The organization of eukaryotic genomes into chromatin structures allows for compaction of the genome and contributes to the regulation of nuclear processes by modulating the accessibility of DNA. The smallest unit of chromatin is the nucleosome core particle, which consists of 147 base pairs (bp) of DNA that wrap 1.65 times around a histone octamer (Luger et al, 1997). Nucleosome core particles are connected by short DNA linkers to form nucleosome arrays. These arrays are often regular, with an approximately constant linker length, and are "phased" when aligned to reference sites, such as active promoters or certain transcription factor binding sites (Baldi et al, 2020). At a larger scale, eukaryotic genomes are organized into chromatin domains. In mammals, these domains include compartments and topologically associating domains (TADs) (Rowley and Corces, 2018). Compartments reflect the separation of euchromatin and heterochromatin (Hildebrand and Dekker, 2020), whereas TADs represent local domains formed by an active process of loop extrusion by cohesin (Fudenberg et al, 2016; Sanborn et al, 2015). Loop extrusion is directed by boundary elements bound to the CCCTC-binding factor (CTCF) via a specific interaction between cohesin and CTCF that causes stalling of extruding cohesin molecules (Li et al, 2020). This allows cohesin to mediate broad interactions between chromatin regions located within, but not beyond, CTCF boundaries, resulting in the formation of relatively insulated domains (TADs), as well as specific interactions (loops) between the CTCF binding sites at TAD borders (Fudenberg et al, 2016; Sanborn et al, 2015).

It is of interest that CTCF binding sites are characterized by prominent phased nucleosome arrays extending up to 20 nucleosomes with a nucleosome repeat length of ~185 bp (Clarkson et al, 2019; Fu et al, 2008). These arrays are dependent on imitation switch (ISWI) chromatin remodelers (Barisic et al, 2019; Bomber et al, 2023; Corin et al, 2025; Iurlaro et al, 2024; Qiu et al, 2015; Tajmul et al, 2025; Wiechens et al, 2016). The relatively stable binding of CTCF to chromatin (Hansen et al, 2017), mediated by its 11 zinc fingers, may contribute to the prominence of these arrays. It is unknown whether the regular pattern of nucleosome organization at CTCF binding sites contributes to the function of CTCF as a TAD insulator protein. It has been shown that regular nucleosome positioning surrounding transcription factor binding sites in *S. cerevisiae* drives the formation of chromatin domains (Fouziya et al, 2024; Hsieh et al, 2015; Oberbeckmann et al, 2024; Wiese et al, 2019), suggesting the possibility that the organization of nucleosomes also influences the structure of chromatin domains in higher eukaryotes. However, the role of nucleosome organization in TAD formation is challenging to address, because it is difficult to separate the contribution of the regular nucleosome arrays at their boundaries from the contribution of CTCF binding levels, as perturbations of CTCF and ISWI remodelers affect both (Barisic et al, 2019; Bomber et al, 2023; Iurlaro et al, 2024; Nora et al, 2017; Wiechens et al, 2016).

[1]Max Planck Institute for Multidisciplinary Sciences, Genome Organization and Regulation, Göttingen, Germany. [2]University of Göttingen, Göttingen, Germany. [3]Max Planck Institute for Multidisciplinary Sciences, Department of Molecular Biology, Göttingen, Germany. [4]Present address: Department of Developmental Biology, Stanford University, Stanford, CA, USA. ✉E-mail: marieke.oudelaar@mpinat.mpg.de

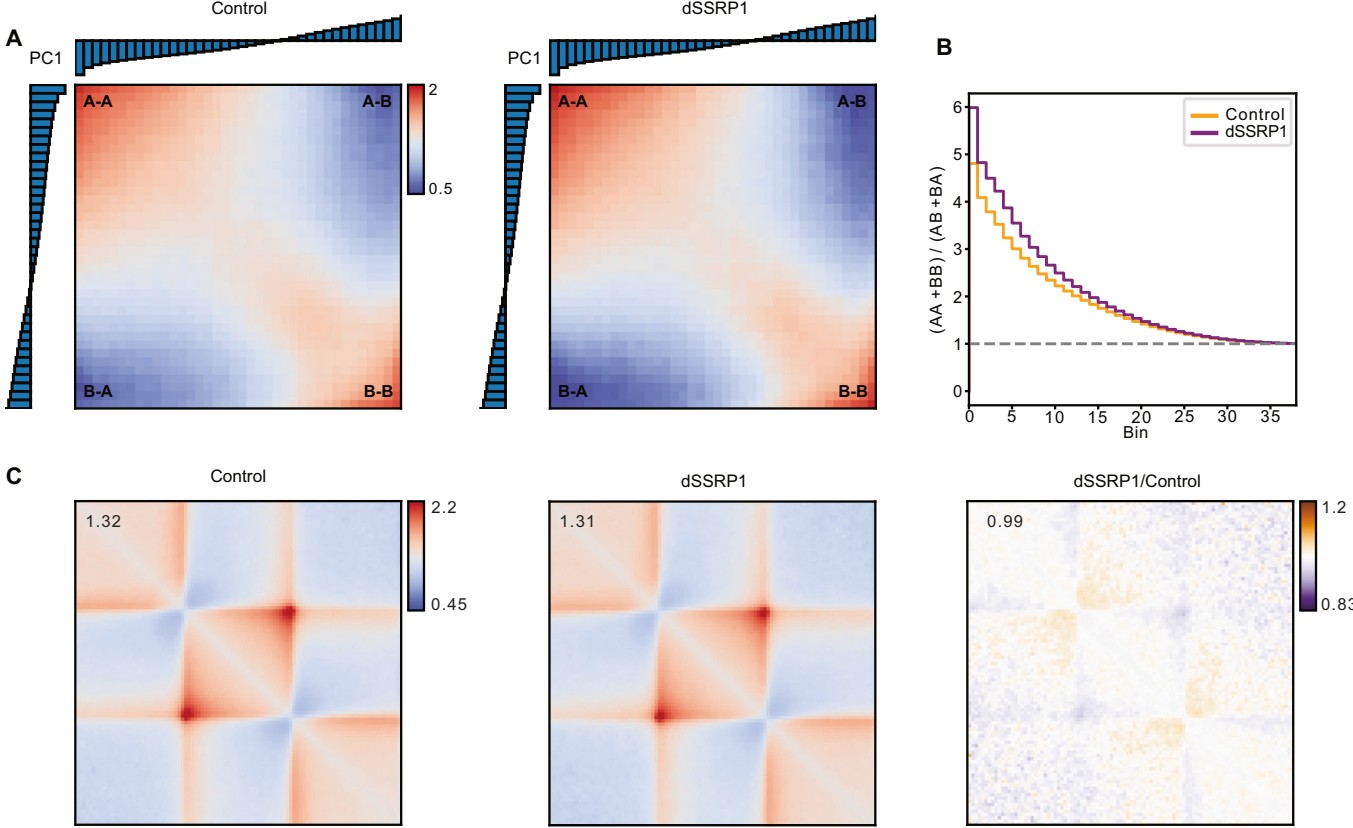

**Figure 1. Rapid depletion of FACT leads to changes in 3D genome organization.**

(A) Saddleplots displaying observed/expected intra- and intercompartment interactions of genomic regions stratified into 39 bins by Eigenvector decomposition in control (left) and SSRP1-depleted (right) K562 cells. Histograms show Eigenvalues of stratified bins. Micro-C data show the average of two biological replicates. (B) Step-plot showing the saddle strength profile (intracompartmental/intercompartmental interactions) in control (orange) and SSRP1-depleted (purple) cells. Data as in (A). (C) Mean observed/expected contact frequencies at TADs ($n = 1645$) in control (left) and SSRP1-depleted (middle) cells and their relative differences (right; purple indicates enriched contacts in control cells; orange indicates enriched contacts in SSRP1-depleted cells). TADs are rescaled to the same size. The average signal is reported in the top left. Data as in (A).

Here, we address this question by studying the role of the histone chaperone facilitates chromatin transcription (FACT) in 3D genome organization. The FACT complex, composed of the subunits structure-specific recognition protein 1 (SSRP1) and suppressor of Ty 16 (SPT16) in human (Orphanides et al, 1999), was originally identified as a factor that stimulates transcription of chromatinized DNA in vitro (Orphanides et al, 1998). Several subsequent in vitro studies have demonstrated a role for FACT in both chromatin transcription and maintenance of chromatin structure in transcribed regions (Jeronimo and Robert, 2022), which is further supported by recent high-resolution structures of the FACT complex (Ehara et al, 2022; Farnung et al, 2021; Liu et al, 2020). For a long time, the precise contribution of FACT to transcription and chromatin structure in vivo remained less clear (Jeronimo and Robert, 2022). Using a human cell line that allows for rapid depletion of the FACT subunit SSRP1, we recently showed that FACT maintains nucleosome organization in transcribed regions and positively regulates transcription elongation in vivo (Žumer et al, 2024). In addition, we identified changes in small-scale chromatin structures after FACT depletion (Žumer et al, 2024). However, since we used a high-resolution chromosome conformation capture (3C) approach targeted at relatively small genomic regions, the effects of

FACT depletion (and the associated disruption of nucleosome organization) on higher-order 3D genome organization at a larger scale remain unclear. To address this, we have performed genome-wide Micro-C experiments after FACT depletion. Integration of Micro-C data with other genomic data shows that FACT depletion perturbs TAD insulation and CTCF loops but does not lead to corresponding changes in CTCF and cohesin occupancy. This therefore demonstrates that regular nucleosome organization as controlled by FACT directly contributes to TAD formation.

## Results

### Rapid depletion of FACT leads to changes in 3D genome organization

We used a K562 SSRP1-dTAG cell line (Žumer et al, 2024) to investigate the role of FACT in the regulation of chromatin structure. Calibrated ChIPmentation experiments show that FACT is efficiently depleted from chromatin in these cells within 4 h of treatment with the dTAG7 ligand (Fig. EV1A). To investigate the

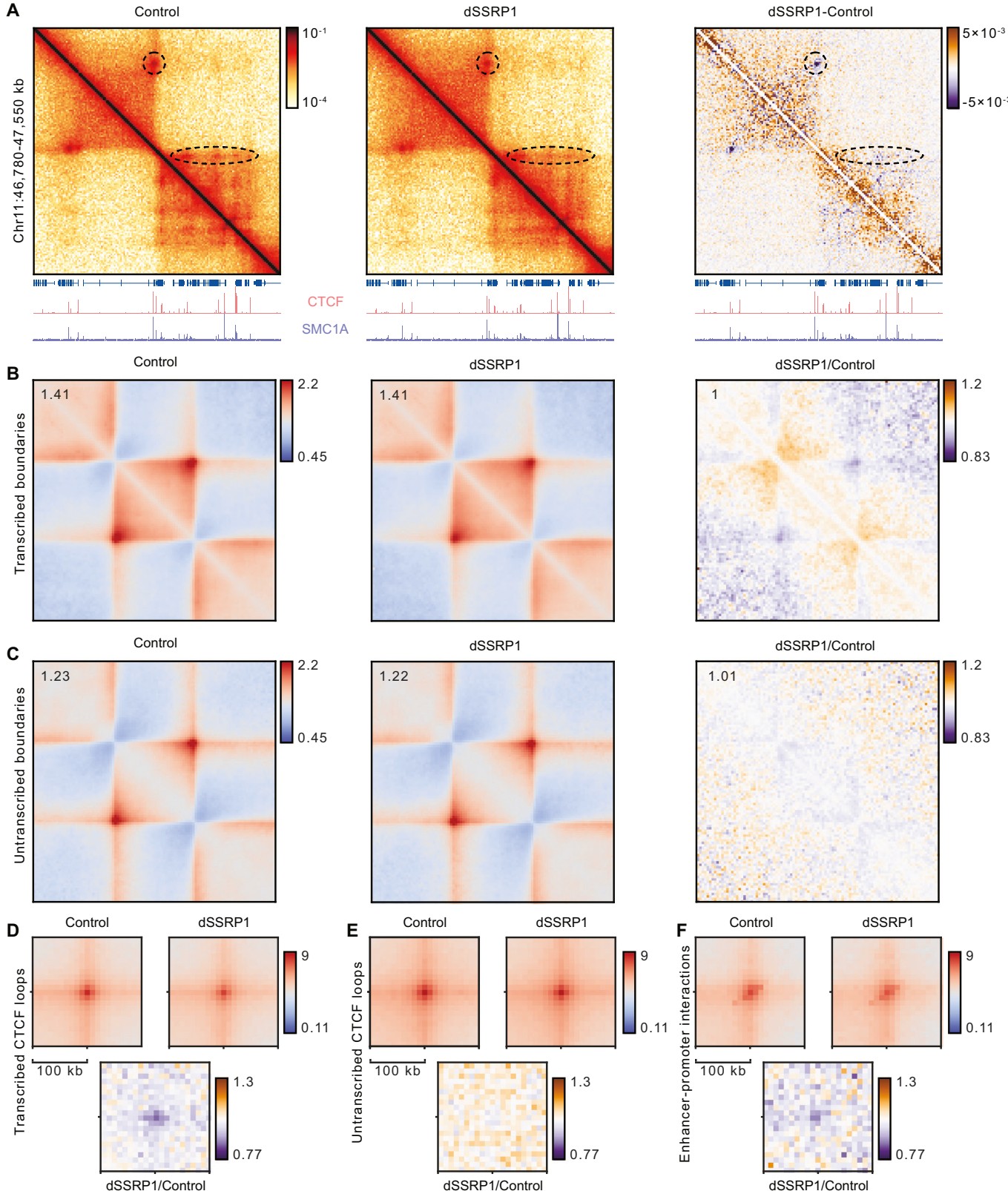

**Figure 2.  FACT depletion weakens TAD insulation and CTCF loops in transcribed regions.**

(A) Micro-C contact matrices (5 kb resolution) of an exemplary region on chromosome 11 in control (left) and SSRP1-depleted (middle) K562 cells and their absolute differences (right; purple indicates enriched contacts in control cells; orange indicates enriched contacts in SSRP1-depleted cells). Dashed highlights indicate regions in which loop strength is reduced. Gene annotation and ChIPmentation tracks for CTCF and SMC1A in control (left and right) and SSRP1-depleted cells (middle) are shown at the bottom. Micro-C data show the average of two biological replicates; ChIPmentation data for CTCF and SMC1A show the average of five and three biological replicates, respectively. (B) Mean observed/expected contact frequencies at TADs of which both boundaries are transcribed ($n = 521$) in control (left) and SSRP1-depleted (middle) cells and their relative differences (right; purple indicates enriched contacts in control cells; orange indicates enriched contacts in SSRP1-depleted cells). TADs are rescaled to the same size. The average signal is reported in the top left. Data as in (A). (C) Same as (B) for TADs of which both boundaries are untranscribed ($n = 477$). (D) Mean observed/expected contact frequencies at CTCF loops of which both anchors are transcribed ($n = 507$) in control (left) and SSRP1-depleted (right) cells and their relative differences (bottom; purple indicates enriched contacts in control cells; orange indicates enriched contacts in SSRP1-depleted cells). Data as in (A). (E) Same as (D) for untranscribed CTCF loop anchors ($n = 789$). (F) Same as (D) for enhancer–promoter interactions ($n = 244$).

effects of FACT depletion on 3D genome organization, we performed Micro-C experiments (Table EV1) (Hsieh et al, 2020; Krietenstein et al, 2020). Analysis of Micro-C interaction frequencies as a function of genomic distance does not reveal major changes upon FACT depletion (Fig. EV1B). However, further analysis shows that depletion of FACT leads to changes in genome compartmentalization, including increased separation between active A and inactive B compartments (Figs. 1A,B and EV1C). Moreover, in the absence of FACT, we observe alterations in TAD structure, involving increased interactions spanning TAD boundaries, indicating a reduction in TAD insulation, as well as weaker CTCF loops (Fig. 1C).

## FACT depletion weakens TAD insulation and CTCF loops in transcribed regions

We previously showed that depletion of FACT leads to a global reduction of transcription (Žumer et al, 2024). To investigate if the changes in 3D genome organization upon FACT depletion can be attributed to changes in transcription, we compared our Micro-C data to a previously published Micro-C dataset in DLD-1 cells after depletion of RNA Polymerase II (RNAPII) (Zhang et al, 2023; Data ref: Zhang et al, 2023). Interestingly, we find that depletion of RNAPII has opposite effects on compartment organization (Fig. EV1D,E) and TAD structure, causing strengthening of TAD insulation and CTCF loops (Fig. EV1F). This is consistent with previous experiments involving chemical inhibition of transcription (Barshad et al, 2023; Hsieh et al, 2020; Ramasamy et al, 2023; Zhang et al, 2023) and indicates that the effects of FACT depletion on large-scale 3D genome organization are not secondary to changes in transcription.

Inspection of regions where we observe altered TAD organization (Figs. 2A and EV2A–C) suggests that the biggest changes occur in transcribed regions. To assess this more systematically, we classified TADs as transcribed or untranscribed, depending on whether their boundaries are located in transcribed or untranscribed regions of the genome. Comparison of transcribed and untranscribed TADs shows that changes in TAD structure are more pronounced in transcribed regions (Fig. 2B,C). These effects are very similar across independent biological replicates (Fig. EV2D). Consistent with the observed changes in TAD structure, we observe a notable reduction in CTCF loops in transcribed regions but not in untranscribed regions (Fig. 2D,E). Further quantification of TAD insulation and CTCF looping confirms that the impairment of these features is more significant in transcribed regions compared to untranscribed regions (Fig. EV2E,F), in line with an active role for FACT in maintaining chromatin structure in transcribed regions. In addition to the impact of FACT depletion on TADs, we also observe a subtle, yet significant decrease in putative

enhancer–promoter interactions, defined as loops with H3K27ac at both anchors (Figs. 2F and EV2F), which is consistent with a previous study (Crump et al, 2023). Since RNAPII depletion causes a marked decrease in enhancer–promoter interactions (Zhang et al, 2023), it is possible that these effects are secondary to reduced transcription levels after FACT depletion.

## CTCF and cohesin occupancy are not reduced following FACT depletion

Since TADs are thought to be formed by a process of cohesin-mediated loop extrusion directed by CTCF boundary elements, a possible explanation for the observed weakening of TADs is that FACT depletion reduces the occupancy of CTCF and/or cohesin on chromatin. To test this, we performed calibrated ChIPmentation experiments for CTCF and the cohesin subunit SMC1A after 4 h of dTAG7 treatment and in control conditions (Fig. 3A–F). Unexpectedly, we observe a minor, yet significant, overall increase in CTCF and cohesin occupancy levels (Fig. EV3). Since this effect is more pronounced in transcribed regions compared to untranscribed regions, the slight increase in CTCF and cohesin occupancy may be explained by increased accessibility for protein binding as a result of reduced nucleosome occupancy in these regions (discussed below). Together, these analyses show that FACT depletion does not lead to a decrease in CTCF or cohesin occupancy that could explain the reduction of TAD insulation and CTCF loops in transcribed regions upon FACT depletion.

Since it remains possible that FACT depletion causes changes in CTCF or cohesin occupancy at specific sites that are masked in a global analysis but contribute to the observed changes in the TAD metaplots, we performed statistical analyses to identify all CTCF and cohesin peaks that are significantly different between the FACT-depleted and control conditions. Next, we plotted TADs with boundaries at which CTCF and cohesin occupancy are not significantly changed after FACT depletion (Fig. 3G,H). This analysis reveals a similar decrease in insulation and looping at transcribed CTCF binding sites as shown in Fig. 2B, indicating that the changes in TAD organization following FACT depletion cannot be attributed to changes in CTCF or cohesin occupancy.

## FACT depletion leads to commensurate changes in nucleosome organization and TAD structure

Several lines of evidence indicate that the formation of chromatin domains in *S. cerevisiae* is driven by regular nucleosome organization (Fouziya et al, 2024; Hsieh et al, 2015; Oberbeckmann

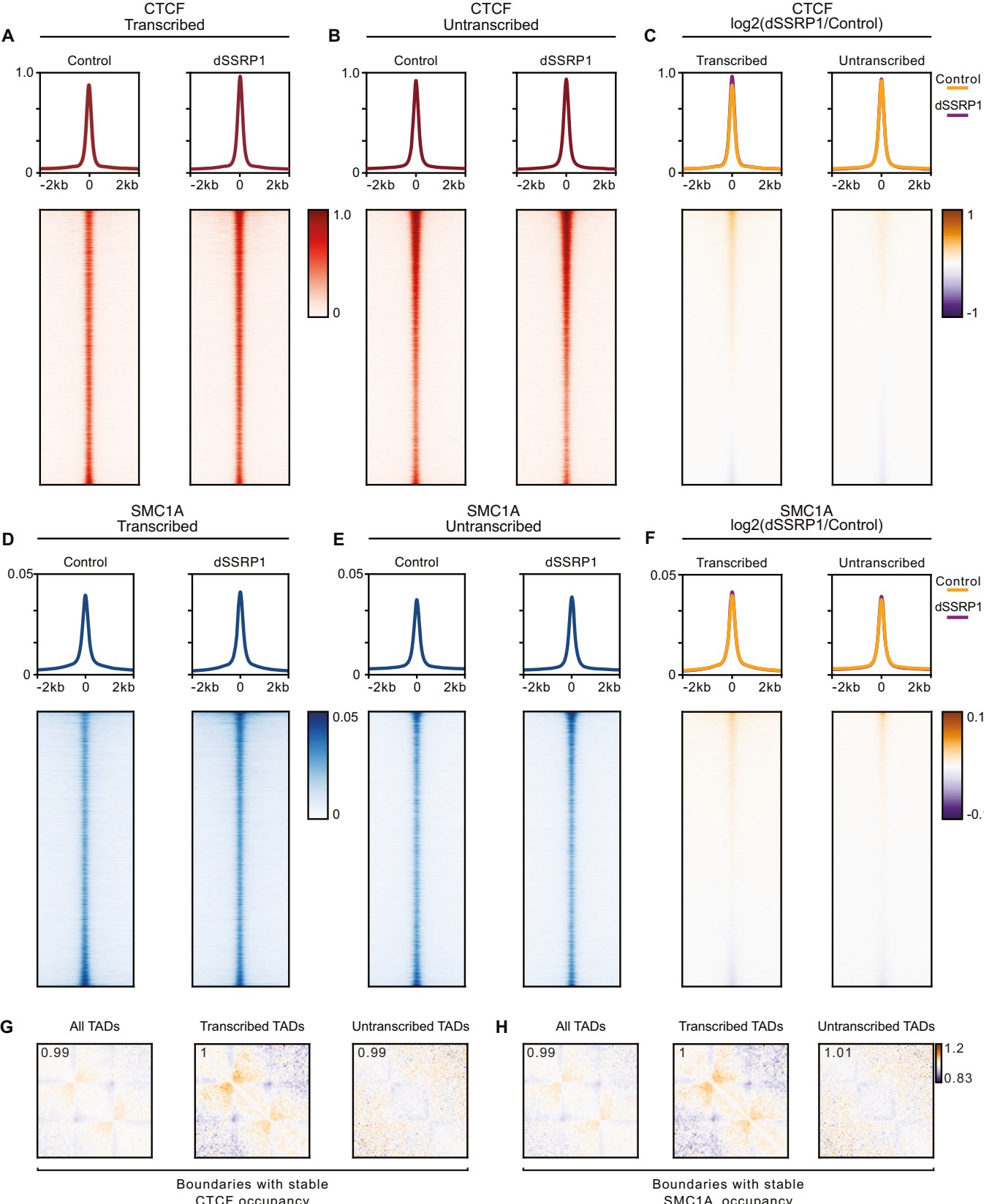

Figure 3.  CTCF and cohesin occupancy are not reduced following FACT depletion.

(A) CTCF ChIPmentation signals at transcribed CTCF binding sites (*n* = 22,681) in control (left) and SSRP1-depleted (right) K562 cells. Heatmaps are sorted based on the differential signals in (C). Data show the average of five biological replicates. (B) Same as (A) for untranscribed CTCF binding sites (*n* = 20,099). (C) Relative differences in CTCF ChIPmentation signals at transcribed (left) and untranscribed (right) CTCF binding sites between control (orange) and SSRP1-depleted (purple) cells. Data as in (A). (D) SMC1A ChIPmentation signals at transcribed CTCF binding sites (*n* = 22,681) in control (left) and SSRP1-depleted (right) cells. Heatmaps are sorted based on the differential signals in (F). Data show the average of three biological replicates. (E) Same as (D) for untranscribed CTCF binding sites (*n* = 20,099). (F) Relative differences in SMC1A ChIPmentation signals at transcribed (left) and untranscribed (right) CTCF binding sites between control (orange) and SSRP1-depleted (purple) cells. Data as in (D). (G) Relative differences in mean observed/expected contact frequencies at all TADs (left; *n* = 1437), TADs with transcribed boundaries (middle; *n* = 396), and TADs with untranscribed boundaries (right; *n* = 473) in control and SSRP1-depleted cells (purple indicates enriched contacts in control cells; orange indicates enriched contacts in SSRP1-depleted cells). Only TADs with boundaries overlapping CTCF peaks that are unchanged following SSRP1 depletion are included. Data as in Fig. 2A. (H) Same as (G) for TADs with boundaries overlapping cohesin peaks that are unchanged following SSRP1 depletion (left: *n* = 1645; middle: *n* = 521; right: *n* = 477).

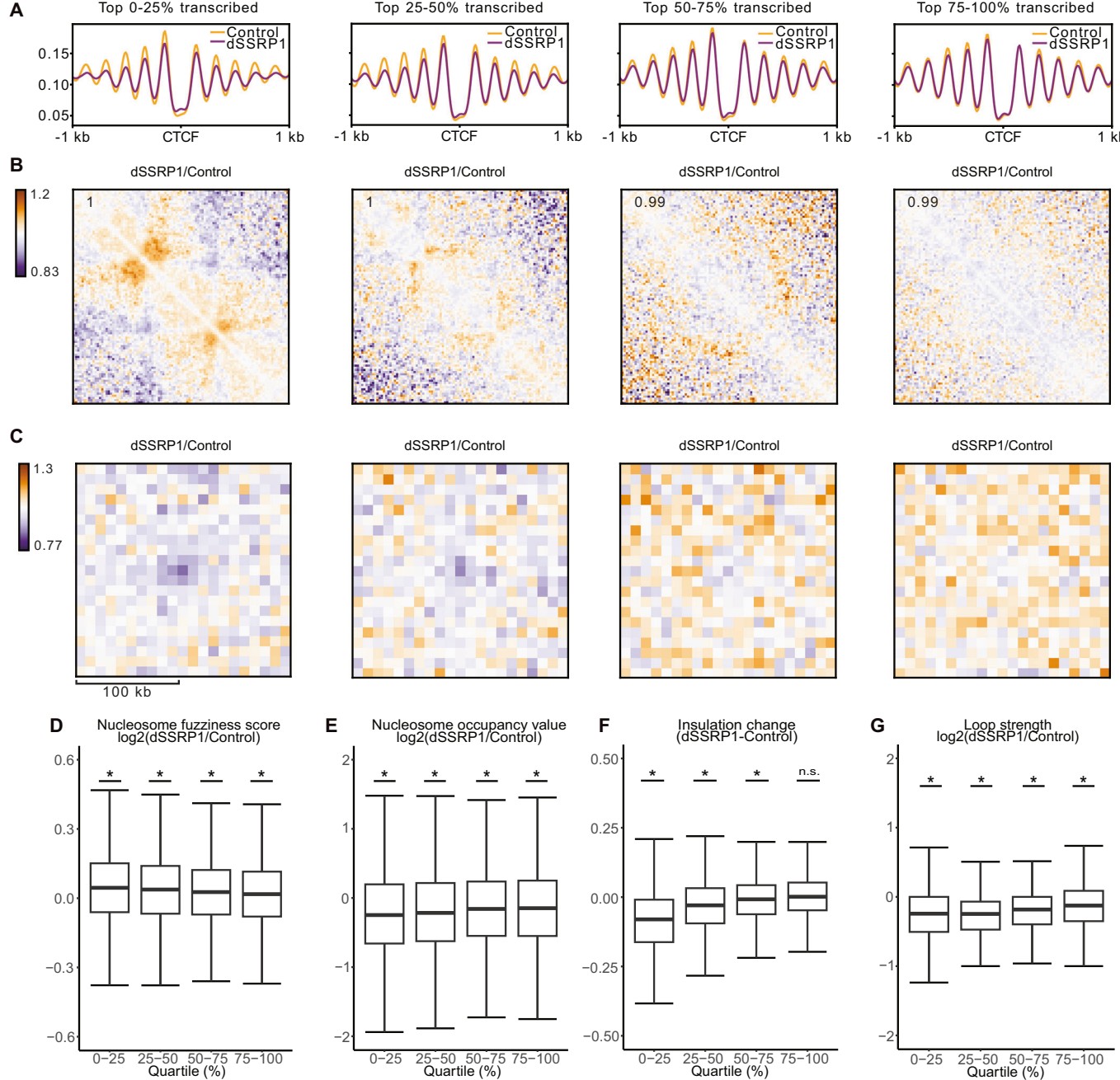

◀ **Figure 4. FACT depletion leads to commensurate changes in nucleosome organization and TAD structure.**

(A) Metaplots showing the mean smoothed dyad signal (derived from MNase-seq data) at TAD boundaries bound by CTCF, categorized into quartiles based on transcriptional activity (0–25%: $n = 1307$; 25–50%: $n = 1306$; 50–75%: $n = 1307$; 75–100%: $n = 1307$), in control (orange) and SSRP1-depleted (purple) K562 cells. Profiles are centered on the midpoint of CTCF binding sites and aligned based on the orientation of the CTCF motifs. Data show the average of two biological replicates. (B) Relative differences in mean observed/expected contact frequencies at TADs of which both boundaries are in the same quartile of transcriptional activity (0–25%: $n = 165$; 25–50%: $n = 85$; 50–75%: $n = 84$; 75–100%: $n = 211$) between control and SSRP1-depleted cells (purple indicates enriched contacts in control cells; orange indicates enriched contacts in SSRP1-depleted cells). TADs are rescaled to the same size. The average signal is reported in the top left. Data as in Fig. 2A. (C) Relative differences in mean observed/expected contact frequencies at CTCF loops of which both anchors are in the same quartile of transcriptional activity (0–25%: $n = 232$; 25–50%: $n = 226$; 50–75%: $n = 251$; 75–100%: $n = 371$) between control and SSRP1-depleted cells (purple indicates enriched contacts in control cells; orange indicates enriched contacts in SSRP1-depleted cells). Data as in Fig. 2A. (D) Relative differences in nucleosome fuzziness at TAD boundaries bound by CTCF, categorized into quartiles based on transcriptional activity (two-sided Wilcoxon rank-sum test; 0–25%: $n = 14,174$, $P < 2.2e$-16; 25–50%: $n = 14,863$, $P < 2.2e$-16; 50–75%: $n = 16,050$, $P < 2.2e$-16), between control and dSSRP1-depleted cells. Boxplots show the interquartile range (IQR) and median of the data; whiskers indicate the minima and maxima within 1.5 * IQR; asterisks indicate statistical significance ($P < 0.05$); n.s. = non-significant. Data as in (A). (E) Same as (D) for nucleosome occupancy (two-sided Wilcoxon rank-sum test; 0–25%: $n = 14,174$, $P < 2.2e$-16; 25–50%: $n = 14,863$, $P < 2.2e$-16; 50–75%: $n = 15,871$, $P < 2.2e$-16; 75–100%: $n = 16,050$, $P < 2.2e$-16). (F) Same as panel D for insulation (two-sided Wilcoxon rank-sum test; 0–25%: $n = 1307$, $P < 2.2e$-16; 25–50%: $n = 1306$, $P < 2.2e$-16; 50–75%: $n = 1307$, $P = 6.856e$-6; 75–100%: $n = 1307$, $P = 0.2831$). Note that the difference is not log-transformed, as the insulation score represents a log-transformed value. Data as in Fig. 2A. (G) Same as (D) for loop strength (two-sided Wilcoxon rank-sum test; 0–25%: $n = 232$, $P < 2.2e$-16; 25–50%: $n = 226$, $P < 2.2e$-16; 50–75%: $n = 251$, $P < 2.2e$-16; 75–100%: $n = 371$, $P < 2.2e$-16). Data as in Fig. 2A.

et al, 2024; Wiese et al, 2019). This suggests that the changes in TAD structure that we observe following FACT depletion may be dependent on changes in the organization of nucleosomes. In line with this hypothesis, MNase-seq data (Žumer et al, 2024; Data ref: Žumer et al, 2024) show that FACT depletion leads to changes in the patterns of nucleosomes at CTCF binding sites (Fig. 4A). To investigate how changes in nucleosome organization relate to changes in TAD structure, we segregated the MNase-seq and Micro-C data into quartiles based on transcription activity (Fig. 4A–C). This analysis shows that nucleosome organization (Fig. 4A), TAD insulation (Fig. 4B), and CTCF loops (Fig. 4C) are most strongly perturbed in highly transcribed regions. Importantly, we observe that the severity of the perturbation of nucleosome organization is predictive of the degree to which TAD insulation and CTCF looping are reduced, consistent with a direct role for nucleosome organization in TAD formation. Quantification of nucleosome fuzziness (Fig. 4D), nucleosome occupancy (Fig. 4E), TAD insulation (Fig. 4F), and CTCF loops (Fig. 4G) across the quartiles further confirms that the changes in these features are commensurate across the quartiles. Of note, many of these features are also significantly changed in the lowest transcribed regions, although the changes in these regions are extremely minor.

The observation that the reduction in TAD insulation is commensurate with changes in nucleosome organization upon FACT depletion indicates that regular nucleosome organization contributes to domain insulation. This predicts that genomic sites that are not bound by CTCF but characterized by regularly spaced and phased nucleosome arrays are also associated with insulation. Such arrays are present at active gene promoters, which are enriched at TAD borders (Dixon et al, 2012; Nora et al, 2012). However, given the interplay between transcription and genome organization, it is difficult to separate the contribution of nucleosome organization and transcription to insulation at active promoters. Binding sites for the restrictive element-1 silencing transcription factor (REST; also called neuron-restrictive silencing factor, NRSF) are also characterized by phased nucleosome arrays (Barisic et al, 2019; Valouev et al, 2011). Consistent with a role for FACT-dependent nucleosome organization in TAD formation, we observe patterns of local insulation at REST binding sites, which also weaken after FACT depletion (Fig. EV4).

## Discussion

Although we have an increasingly detailed understanding of the different layers of genome organization, the interplay between chromatin structures across scales remains unclear (Aljahani et al, 2024). Orthogonal lines of evidence, based on computational simulations (Wiese et al, 2019), in vitro reconstitution of chromatin domains (Oberbeckmann and Oudelaar, 2024), and combinatorial perturbations of chromatin remodelers (Fouziya et al, 2024), indicate that regular nucleosome positioning drives the formation of interphase chromatin domains in *S. cerevisiae*, which range between ~2 and 10 kb in size. However, it remains unclear how nucleosome organization influences the formation of much larger-scale domains in mammals, including TADs, which are usually between 100 kb and 1 Mb in size. This question is not straightforward to address, as perturbation of nucleosome organization generally also affects the binding of proteins that regulate genome folding, including CTCF. For example, it has been shown that deletion of the ISWI subunit SNF2H weakens TAD insulation and CTCF loops (Barisic et al, 2019), but since this perturbation also reduces CTCF binding, it is difficult to assess the direct contribution of chromatin remodeling and nucleosome organization to TAD formation. Here, we show that depletion of FACT impacts the organization of nucleosomes at CTCF binding sites without reducing CTCF binding levels, which therefore allows us to separate the role of nucleosome organization and CTCF binding in the formation of TADs.

It is important to consider that the changes in 3D genome organization following FACT depletion could in principle be secondary to changes in transcription, since it has been shown that FACT depletion leads to a global reduction in transcription levels (Žumer et al, 2024) and that the process of transcription influences 3D genome organization (Banigan et al, 2023; Barshad et al, 2023; Brandão et al, 2019; Busslinger et al, 2017; Hsieh et al, 2020; Zhang et al, 2023; Zhang et al, 2021). However, Micro-C data in RNAPII-depleted human cells show the opposite effects on TADs, including stronger TAD insulation and CTCF loops (Zhang et al, 2023). Since this is associated with commensurate changes in cohesin occupancy, these results have been interpreted as antagonism between transcription and loop extrusion, which would explain the strengthening of TADs in the absence of RNAPII. We do not observe this in our data,

likely because transcription is much less affected by 4 h of FACT depletion compared to 16 h of RNAPII depletion.

Since the observed changes in TAD structure following FACT depletion cannot be explained by changes in protein occupancy or transcription, we can only attribute them to the perturbed organization of nucleosomes at CTCF binding sites upon FACT depletion, which occurs predominantly in the transcribed regions where we observe a commensurate weakening of TAD insulation and CTCF loops. The changes in nucleosome organization include a significant increase in nucleosome fuzziness and a significant decrease in nucleosome occupancy, in line with previously reported loss of histones following FACT perturbation (Jeronimo and Robert, 2022). Interestingly, our findings are consistent with a recent preprint that shows that reduced nucleosome density after knock-out of the H3.3-specific chaperone HIRA is associated with reduced insulation at active genes (Karagyozova et al, 2024). Of note, it is possible that the observed strengthening of TAD insulation and CTCF loops following RNAPII depletion (Zhang et al, 2023) is also (in part) related to changes in nucleosome organization, since active transcription perturbs regular organization of nucleosomes (Weiner et al, 2010). Loss of transcription may therefore allow for the formation of more regular and stable nucleosome conformations at CTCF binding sites, which may result in stronger insulation and CTCF loops. In line with this hypothesis, we have previously observed an increase in nucleosome regularity and insulation at intragenic CTCF binding sites in the context of reduced transcription following Mediator depletion (Ramasamy et al, 2023).

What are the molecular mechanisms by which FACT-dependent nucleosome organization contributes to TAD insulation and CTCF looping? It is possible that domain formation and/or insulation are facilitated in the context of regular nucleosome arrays because regularly positioned nucleosomes promote intra-fiber nucleosome interactions, whereas irregularly positioned nucleosomes are more likely to engage in inter-fiber interactions (Jentink et al, 2023; Moore et al, 2025). Furthermore, regular nucleosome organization may contribute to the stability of CTCF loops through interdigitation of nucleosomes in the regular arrays surrounding the two CTCF binding sites that are engaged in the loop (Adhireksan et al, 2020). Through such mechanisms, the arrangement of nucleosomes can influence properties of the chromatin fiber that contribute to the formation and insulation of chromatin domains and loops. The observation that local insulation can be observed at binding sites for the transcription factor REST, which has not been reported to interact with cohesin, and our previous observation that local insulation around CTCF binding sites is not affected by cohesin depletion (Aljahani et al, 2022), suggest that nucleosome organization contributes to domain insulation in a loop-extrusion-independent manner.

However, it is possible that the organization of nucleosomes also contributes to higher-order genome organization by influencing cohesin-mediated loop extrusion. Even though we do not observe clear changes in the occupancy patterns of cohesin, it is possible that our data are not sensitive enough to detect dynamic changes in the extrusion trajectory of cohesin that may be caused by FACT-dependent changes in nucleosome organization. In line with this hypothesis, it has been shown that FACT contributes to the positioning of cohesin on chromatin in yeast, possibly by stabilizing cohesin on chromatin and/or facilitating its movement through nucleosome arrays (Garcia-Luis et al, 2019). It has also been proposed that perturbed nucleosome organization may impede loop extrusion by reducing the compaction of chromatin fibers and thereby increasing the

effective distance that cohesin molecules travel on chromatin (Karagyozova et al, 2024; Maji et al, 2020). Furthermore, it is of interest that previous studies based on deletion of the ISWI subunit BPTF (Iurlaro et al, 2024) and on analysis of specific disease-associated CTCF mutations (Do et al, 2025) suggest that the accessibility of chromatin influences cohesin functioning. In line with this, a recent preprint shows that the accessibility of TAD borders correlates with insulation strength (Jácome-López et al, 2024). Together, these findings suggest that nucleosome organization and the resulting properties of chromatin fibers might influence cohesin-mediated loop extrusion, although their precise interplay and the underlying molecular mechanisms remain unclear.

Although the impact of FACT depletion on TAD structure is modest, with a reduction in insulation and looping of ~20% in highly transcribed regions, it is important to consider that the effects of FACT depletion on nucleosome organization at CTCF binding sites are also modest. It is therefore difficult to estimate the overall importance of nucleosome organization for TAD formation based on the FACT depletion data alone. However, our work shows that the organization of nucleosomes at CTCF binding sites contributes to TAD insulation and CTCF looping independently of CTCF binding. The overall contribution of nucleosome organization to higher-order chromatin folding, as well as the underlying molecular mechanisms and potential interplay with loop extrusion, are exciting areas to explore in more detail in future research.

# Methods

**Reagents and tools table**

| Reagent/resource | Reference or source | Identifier or catalog number |
|---|---|---|
| **Experimental models** | | |
| K562 SSRP1-dTAG cell line (clone N1C10; H. sapiens) | Žumer et al, 2024 | N/A |
| S2 cell line (D. melanogaster) | DSMZ | ATCC Cat#CRL-1963, RRID:CVCL_Z232 |
| **Recombinant DNA** | | |
| **Antibodies** | | |
| Anti-HA | Roche | Cat#11867423001, RRID:AB390918 |
| Anti-CTCF | Diagenode | Cat#C15410210, RRID:AB_2753160 |
| Anti-SMC1A | Abcam | Cat#ab9262, RRID:AB_307121 |
| Spike-in antibody | Active Motif | Cat#61686, RRID:AB_2737370 |
| **Oligonucleotides and other sequence-based reagents** | | |
| **Chemicals, enzymes, and other reagents** | | |
| dTAG7 ligand | Tocris | Cat#6912 |
| DMSO | Sigma-Aldrich | Cat#D8418-50ML |
| DSG | Thermo Fisher Scientific | Cat#20593 |
| MNase | New England Biolabs | Cat#M0247 |
| ATP | Jena Bioscience | Cat#NU-1010 |

| Reagent/resource | Reference or source | Identifier or catalog number |
|---|---|---|
| DTT | Roche | Cat#10197777001 |
| Klenow fragment | New England Biolabs | Cat#M0210L |
| Biotin-14-dATP | Jena Bioscience | Cat#NU-835BIO14-L |
| Biotin-11-dCTP | Jena Bioscience | Cat#NU-809X-L |
| dGTP | Jena Bioscience | Cat#NU-1003L |
| dTTP | Jena Bioscience | Cat#NU-1004L |
| T4 DNA ligase | New England Biolabs | Cat#M0202L |
| Exonuclease III | New England Biolabs | Cat#M0206L |
| Mag-Bind TotalPure NGS beads | Omega Bio-tek | Cat#M1378-01 |
| Dynabeads MyOne Streptavidin T1 beads | New England Biolabs | Cat#65601 |
| Dynabeads Protein A for Immunoprecipitation | Invitrogen | Cat#10008D |
| Dynabeads Protein G for Immunoprecipitation | Invitrogen | Cat#10003D |
| Illumina Tagment DNA Enzyme | Illumina | Cat#20034197 |
| NEBNest High-Fidelty PCR Master Mix | New England Biolabs | Cat#M0541 |
| **Software** | | |
| nf-core/hic (v.2.1.0) | https://nf-co.re/hic/2.1.0/ Servant et al, 2023 | |
| FastQC (v.0.11.9) | https://www.bioinformatics.-babraham.ac.uk/projects/fastqc/ | |
| Trim Galore! (v.0.6.10) | https://www.bioinformatics.-babraham.ac.uk/projects/trim_galore/ | |
| bowtie2 (v.2.4.4) | https://bowtie-bio.sourceforge.net/bowtie2/ Langmead and Salzberg, 2012 | |
| cooler (v.0.8.11) | https://github.com/open2c/cooler Abdennur and Mirny, 2020 | |
| cooltools (v.0.6.1) | https://github.com/open2c/cooltools Open2C | |
| bioframe (v.0.6.1) | https://github.com/open2c/bioframe Open2C | |
| GenoSTAN | Zacher et al, 2017 | |
| mustache (v.1.0.1) | https://github.com/ay-lab/mustache Roayaei Ardakany et al, 2020 | |
| coolpup.py (v.1.1.0) | https://github.com/open2c/coolpuppy Flyamer et al, 2020 | |

| Reagent/resource | Reference or source | Identifier or catalog number |
|---|---|---|
| deepTools (v.3.5.5) | https://github.com/deeptools/deepTools Ramírez et al, 2016 | |
| MACS2 (v.2.1.2) | https://github.com/macs3-project/MACS Zhang et al, 2008 | |
| DESeq2 (v.1.36.0) | https://bioconductor.org/packages/release/bioc/html/DESeq2.html Love et al, 2014 | |
| **Other** | | |
| DNeasy blood and tissue kit | Qiagen | Cat#69504 |
| NEBNext Ultra II DNA Library Prep Kit for Illumina | New England Biolabs | Cat#E7645 |
| Illumina NextSeq 2000 | Illumina | |

## Cell culture

K562 SSRP1-dTAG cells (Žumer et al, 2024) were cultured in RPMI medium (Thermo Fisher Scientific) supplemented with 10% fetal bovine serum (FBS, Thermo Fisher Scientific) and 1×GlutaMax (Thermo Fisher Scientific). Cultures were maintained at 37 °C with 5% $CO_2$, with cell densities between $1 \times 10^5$ and $7 \times 10^5$ cells/mL. Quarterly mycoplasma testing using the PlasmoTest kit (Invivo-Gen) confirmed that the cells were free of mycoplasma contamination. The cells were not recently authenticated. SSRP1 depletion was performed by adding 500 nM dTAG7 ligand in DMSO to the medium. Cells treated with DMSO only served as controls. Both treatments were performed at a 1:20,000 DMSO dilution for 4 h. Blinding is not applicable to our study. Cells that were fixed and processed as independent aliquots were treated as biological replicates; cell aliquots that were divided into multiple batches during an experimental procedure were treated as technical replicates. All experiments were performed in two to five biological replicates, in line with the standard in the field. *D. Melanogaster* Schneider-2 (S2) cells, used as spike-ins for ChIPmentation experiments, were cultured in Schneider's Drosophila-Medium (Gibco) supplemented with 10% FBS (Gibco) in a non-humidified incubator at 27 °C, protected from light.

## Micro-C

Micro-C experiments were performed as previously described (Goel et al, 2023) in two biological and three technical replicates per condition. In brief, $30 \times 10^6$ cells per replicate were divided into six reactions each and crosslinked in phosphate-buffered saline (PBS, Thermo Fisher Scientific) supplemented with 3 nM disuccinimidyl glutarate (DSG, Thermo Fisher Scientific) and 1% formaldehyde (Thermo Fisher Scientific). The reactions were quenched with 0.375 M Tris-HCl pH 7.5 (Thermo Fisher Scientific). For the following steps, all reactions were consecutively performed in the same tube. To prepare the samples for digestion,

they were lysed in buffer cMB#1 (50 mM NaCl, 10 mM Tris, 5 mM MgCl$_2$, 1 mM CaCl2, 0.2% NP-40, 1×Protease Inhibitor Cocktail (PIC)) at 4 °C. Subsequent digestion with 34 Kunitz U of Micrococcal nuclease (MNase, NEB, M0247) per $1 \times 10^6$ cells was performed for 20 min at 37 °C. 4 mM ethylene glycol-bis(2-aminoethylether)-N,N,N′,N′-tetraacetic acid (EGTA, Sigma-Aldrich, E3889) was added to quench the reaction. A 5% aliquot was assessed for digestion efficiency by de-crosslinking and extracting DNA using DNeasy blood and tissue kit (Qiagen). Samples with 80-90% mono-nucleosomal fragments were selected for further processing. For end-chewing, nuclei were subjected to 1×NEBuffer 2.1 (NEB), 2 mM adenosine 5'-triphosphate (ATP, Jena Bioscience), 5 mM 1,4-dithiothreitol (DTT, Roche), and 0.1 U/ µL T4 Polynucleotide Kinase (T4 PNK, NEB, M0201L) for 15 min at 37 °C with mixing. 0.05 U/µL Klenow fragment (NEB, M0210L) was added and incubated for 15 min at 37 °C with mixing. Fragment ends were labeled with biotin by adding 66 µM Biotin-14-dATP (Jena Bioscience), 66 µM Biotin-11-dCTP (Jena Bioscience), 66 µM ddGTP (Jena Bioscience), 66 µM ddTTP (Jena Bioscience), 1×T4 DNA ligase buffer (NEB), and 100 µg/mL bovine serum albumin (BSA, Sigma-Aldrich). The reaction was incubated for 45 min at 25 °C with interval mixing and quenched with 30 mM ethylenediaminetetraacetic acid (EDTA, Sigma-Aldrich). For proximity ligation, biotinylated samples were incubated with 1×T4 DNA ligase buffer (NEB), 100 µg/mL BSA (Sigma-Aldrich), and 20 U/µL T4 DNA ligase (NEB, M0202L) for 2.5 h at 25 °C. Subsequently, biotin-dNTPs from unligated ends were removed with 1×NEBuffer#1 (NEB) and 5 U/µL Exonuclease III (NEB, M0206L). The ligated product was de-crosslinked and purified using DNeasy blood and tissue kit (Qiagen, 69504). Di-nucleosomal fragments (~360 bp) were selected by two-sided size selection using Mag-Bind TotalPure NGS beads (Omega Bio-tek). After pulldown of biotinylated fragments with Dynabeads MyOne Streptavidin T1 beads (NEB), the samples were indexed "on-bead" using NEBNext Ultra II DNA Library Prep Kit for Illumina (NEB, E7645). Pooled libraries were sequenced on the Illumina NextSeq 2000 platform in 2×66 bp paired-end mode.

## Micro-C data analysis

Micro-C data in DLD-1 cells (Zhang et al, 2023; Data ref: Zhang et al, 2023) were first trimmed to 66 bp read length. Both Micro-C datasets (in DLD-1 and K562 cells) were processed using the nf-core/hic pipeline (v.2.1.0, https://nf-co.re/hic) (Servant et al, 2023) with default options except: --dnase --min_cis_dist 150 --min_mapq 30 --bin_size 100, 150, 200, 500, 1000, 2500, 5000, 10,000 --skip_dist_decay --skip_tads --skip_compartments --skip_mcool. In brief, quality of sequencing reads was assessed using FastQC (v.0.11.9). Reads were mapped to the human genome hg38 using bowtie2 (Langmead and Salzberg, 2012) (v.2.4.4). After removing reads with a mapq-score smaller than 30, detection of valid interaction products and removal of duplicates were performed as specified by the HiC-Pro pipeline (Servant et al, 2015). Using cooler (Abdennur and Mirny, 2020) (v.0.8.11), contact maps were generated at specified resolutions and normalized. As the replicates were consistent across conditions, for final visualizations and downstream analyses, merged datasets were used. For compartment analysis, cooltools (Open2C et al, 2024a) (v0.6.1) was used, calling the eigs-cis function for eigenvector decomposition at 100 kb resolution and bioframe frac_gc to generate

a GC content phasing track. For saddle strength analysis, expected cis interactions were calculated with cooltools expected_cis and corresponding saddles with cooltools saddle. Saddle strengths were calculated with cooltools saddle_strength and saddleplots were created using cooltools saddleplot. Expected cis interactions were also used for distance decay analysis. TAD boundaries were defined based on the default Li thresholded boundaries called using cooltools insulation and overlap with a CTCF binding site (based on ChIPmentation peaks, as described in the MNase-seq analysis section). For TAD annotation, adjacent boundaries were merged with bioframe (Open2C et al, 2024b) (v.0.6.1) and putative TADs exceeding 1.5 Mb in length were excluded. Using TT-seq data (Schwalb et al, 2016) from K562 SSRP1-dTAG cells (Žumer et al, 2024; Data ref: Žumer et al, 2024) and the R/Bioconductor package GenoSTAN (Zacher et al, 2017), the genome was segmented into transcribed and untranscribed states as previously described (Lidschreiber et al, 2021). 78357 transcribed units were identified and defined as transcribed regions. Untranscribed regions were defined as genomic regions that are at least 10 kb away from annotated transcribed units and do not overlap with annotated features in the reference genome. TADs with both boundaries overlapping transcribed CTCF binding site were defined as transcribed TADs; TADs with both boundaries overlapping untranscribed CTCF binding sites were defined as untranscribed. Loops were identified at 10 kb resolution using mustache (Roayaei Ardakany et al, 2020) (v.1.0.1), with options: -d 10,000,000, -pt 0.05, and -st 0.8. Loops with both anchors overlapping at least one CTCF binding site were considered CTCF loops, while loops with both anchors overlapping the H3K27ac histone mark and no or one anchor overlapping a CTCF binding site were considered enhancer–promoter interactions. TADs and CTCF loops were categorized by transcriptional activity by sorting TAD boundaries and CTCF loop anchors into quartiles based on previously published TT-seq data (Žumer et al, 2024; Data ref: Žumer et al, 2024). TADs and loops were assigned to a quartile if both boundaries or anchors fell within the same group. Differential boundary strength was assessed by subtracting the log-transformed boundary strength values (assessed with the cooltools insulation function) in control and dSSRP1 conditions. Loop strength differences were calculated by dividing the contact probabilities of loop anchors between control and dSSRP1 and log-transforming the result. Two-sided Wilcoxon signed-rank tests were performed to assess the statistical significance of changes in insulation and loop strength after visual assessment of normal distribution and variance homogeneity between samples. Pileups were created using coolpup.py (Flyamer et al, 2020) (v.1.1.0) at 10 kb resolution, using expected normalization with a flank of 6 kb for pileups around CTCF and REST binding sites and 100 kb around loops. Pileups of TADs were rescaled to the same size.

## ChIPmentation

ChIPmentation experiments for SSRP1-HA, CTCF, and SMC1A were performed as previously described (Karpinska et al, 2025; Schmidl et al, 2015) in two, five, and three biological replicates, respectively, per condition. In brief, $1.5 \times 10^6$ cells were crosslinked with 1% formaldehyde (Thermo Scientific) for 10 min at room temperature, followed by quenching with 125 mM ice-cold glycine (Sigma-Aldrich) for 5 min and two washes with ice-cold PBS. Nuclei isolation was performed with Farnham lysis buffer (5 mM PIPES pH 8, 85 mM KCl, 0.5% NP-40) and followed by nuclear lysis with 0.5% SDS buffer (10 mM Tris-HCl pH 8, 1 mM EDTA, 0.5% SDS). For CTCF

ChIPmentation, chromatin fragmentation was performed by sonication for 7 min with the following settings on the S220 Covaris sonicator: duty factor: 5%, peak incident power: 140 W, cycles per burst: 200). For SSRP1-HA and SMC1A ChIPmentation, chromatin was first digested by a titrated amount of MNase (2.5 kunitz; NEB, M0247S) at 37 °C for 10 min and followed by gentle sonication (duty factor: 2%, peak incident power: 105 W, cycles per burst: 200) for 1 or 12 min, respectively. The supernatant of the sheared chromatin was mixed with IP buffer (10 mM Tris-HCl pH 8, 1 mM EDTA, 150 mM NaCl, 1% Triton-X-100, 1×PIC). Protein A and Protein G dynabeads (Invitrogen) were mixed in a 1:1 ratio and after three washes with 0.5% BSA blocking solution (Sigma-Aldrich), the mixture was incubated with 2 µg of primary antibody (HA: Cell Signaling, 3724S; CTCF: Diagenode, C15410210-50; SMC1A: Abcam, ab9262) and 1 µg of spike-in antibody (Active Motif, 61686) for 6 h at 4 °C. Sheared chromatin was then added with 200 ng of Drosophila spike-in chromatin to the bead-bound antibodies for 16 h at 4 °C. Post-immunoprecipitation washes were performed using buffers with varying salt concentrations at 4 °C. Sequencing adapters were added using "on-bead" tagmentation using the Illumina Tagment DNA Enzyme (Illumina, 20034197), after which overnight de-crosslinking was performed at 65 °C with proteinase K at a final concentration of 0.2 mg/mL. DNA extraction was performed with 1.8 volumes of Mag-Bind TotalPure NGS beads (Omega Bio-Tek). Library preparation was performed using the NEBNext High-Fidelity 2×PCR Master Mix (NEB, M0541) and two rounds of size selection with 1.2 volumes of Mag-Bind TotalPure NGS beads were performed. Pooled libraries were sequenced on the Illumina NextSeq 2000 platform in 2 × 66 bp paired-end mode.

## ChIPmentation analysis

ChIPmentation data were first trimmed to remove adapter sequences using Trim Galore (v.0.6.10) and then mapped to human genome hg38 and *D. melanogaster* genome dm6 with Bowtie2 (Langmead and Salzberg, 2012) (v.2.3.4.1). Duplicates were marked with Picard tools (v.2.20.2). After removal of duplicated reads and reads mapped to blacklisted regions of the human genome (Amemiya et al, 2019), bigWig coverage tracks were generated with deepTools (Ramírez et al, 2016) (v.3.5.5) bam-Coverage function, using a binning size of 10 bp. All coverage tracks were normalized using spike-in read counts, defined by the total number of properly paired, deduplicated reads mapped to the *D. melanogaster* genome. Coverage tracks of replicates were merged using deepTools bigwigAverage function. Peaks in ChIPmentation data were identified using MACS2 (Zhang et al, 2008) (v.2.1.2) in narrow peak calling mode with a *q* value of 0.01. Consensus peaks from replicates in each condition (control and SSRP1-depleted cells) were retained and then merged to create a unified peak set. To generate metaplots and heatmaps of CTCF and SMC1A ChIPmentation data, CTCF peaks from the unified peak set were first categorized as transcribed or untranscribed as described above. Next, the computeMatrix and plotHeatmap functions from deepTools were used to create metaplots and heatmaps of coverage values associated with CTCF binding sites in transcribed and untranscribed regions. For SSRP1-HA ChIPmentation data, the metaplots and heatmaps were generated based on all SSRP1 peaks identified in the ChIPmentation data. To assess the overall changes in CTCF and SMC1A occupancy upon SSRP1 depletion,

differential coverage tracks were generated using the deepTools bigwigCompare function. Based on the differential coverage tracks, differential metaplots and heatmaps were created. Log-transformed values from the differential coverage tracks were extracted using the deepTools multiBigwigSummary function and visualized as boxplots. Two-sided Wilcoxon signed-rank tests were performed to assess the statistical significance of changes in occupancy. For the differential binding analysis of CTCF and SMC1A ChIPmentation data, the deepTools multiBamSummary function was used to calculate the raw read counts at all CTCF peaks for each replicate. The raw read counts were then converted to an input matrix for DESeq2 (Love et al, 2014) (v.1.36.0). Size factors in DESeq2 were determined using spike-in read counts, and differential binding analysis was conducted with an adjusted *P* value threshold of <0.01.

## MNase-seq analysis

MNase-seq data (Žumer et al, 2024; Data ref: Žumer et al, 2024) were pre-processed as described previously (Žumer et al, 2024). To analyze nucleosome organization at CTCF binding sites across genomic regions, CTCF ChIPmentation peaks were intersected with publicly available CTCF ChIP-seq peaks (ENCODE accession number ENCFF396BZQ (Dunham et al, 2012)) to obtain a set of high-confidence CTCF peaks that are both accurately centered and confirmed in K562 SSRP1-dTAG cells. These peaks were divided into quartiles based on transcriptional activity, as determined by TT-seq data. For each quartile, MNase-seq metaplots were generated with the deepTools computeMatrix and plotProfiles functions. Nucleosome fuzziness and occupancy changes upon SSRP1 depletion were analyzed using the DANPOS tool (Chen et al, 2013) in comparison mode. Log2 fold changes in fuzziness and occupancy of nucleosomes within the range of ±1 kb of quartile-segregated CTCF peaks were extracted from the DANPOS output and visualized as boxplots. Two-sided Wilcoxon signed-rank tests were performed to assess the statistical significance of changes after visual assessment of normal distribution and variance homogeneity between samples.

# Data availability

The datasets produced in this study are available in the following databases: Micro-C data: Gene Expression Omnibus GSE284561. ChIPmentation data: Gene Expression Omnibus GSE284561.

The source data of this paper are collected in the following database record: biostudies:S-SCDT-10_1038-S44320-025-00165-7.

# Peer review information

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

## Acknowledgements

We would like to thank P Cramer for infrastructure support; I Flyamer for support with Micro-C analysis; G Barshad and C Danko for sharing functional

enhancer annotations in K562 cells; M Lidschreiber, G Narlikar, A Papantonis, V Ramani, D Schübeler, and J Söding for helpful discussions; K Maier, M Rohm, P Rus, L Siegmund, and R Walz for experimental support; and all members of the Oudelaar group for discussions and feedback. This work was supported by the Max Planck Society (AMO); the European Research Council (Starter Grant 3D-REG 101115401, AMO); the Deutsche Forschungsgemeinschaft (DFG) via SFB 1565 (project 469281184/P02, AMO); the PhD program "Genome Science"—International Max Planck Research School at the Georg August University Göttingen (CM, AA, and SR); and the MSc/PhD program "Molecular Biology"— International Max Planck Research School at the Georg August University Göttingen (YZ and SS).

## Author contributions

**Clemens Mauksch**: Conceptualization; Formal analysis; Investigation; Visualization; Writing—original draft; Writing—review and editing. **Yi Zhu**: Conceptualization; Formal analysis; Visualization; Writing—original draft; Writing—review and editing. **Taras Velychko**: Conceptualization; Investigation; Writing—review and editing. **Spyridoula Sagropoulou**: Conceptualization; Investigation; Writing—review and editing. **Abrar Aljahani**: Conceptualization; Investigation; Writing—review and editing. **Shyam Ramasamy**: Conceptualization; Supervision; Writing—review and editing. **Kristina Žumer**: Conceptualization; Resources; Supervision; Writing—review and editing. **A Marieke Oudelaar**: Conceptualization; Supervision; Funding acquisition; Visualization; Writing—original draft; Project administration; Writing—review and editing.

Source data underlying figure panels in this paper may have individual authorship assigned. Where available, figure panel/source data authorship is listed in the following database record: biostudies:S-SCDT-10_1038-S44320-025-00165-7.

## Funding

## Disclosure and competing interests statement

The authors declare no competing interests.

# Expanded View Figures/Table

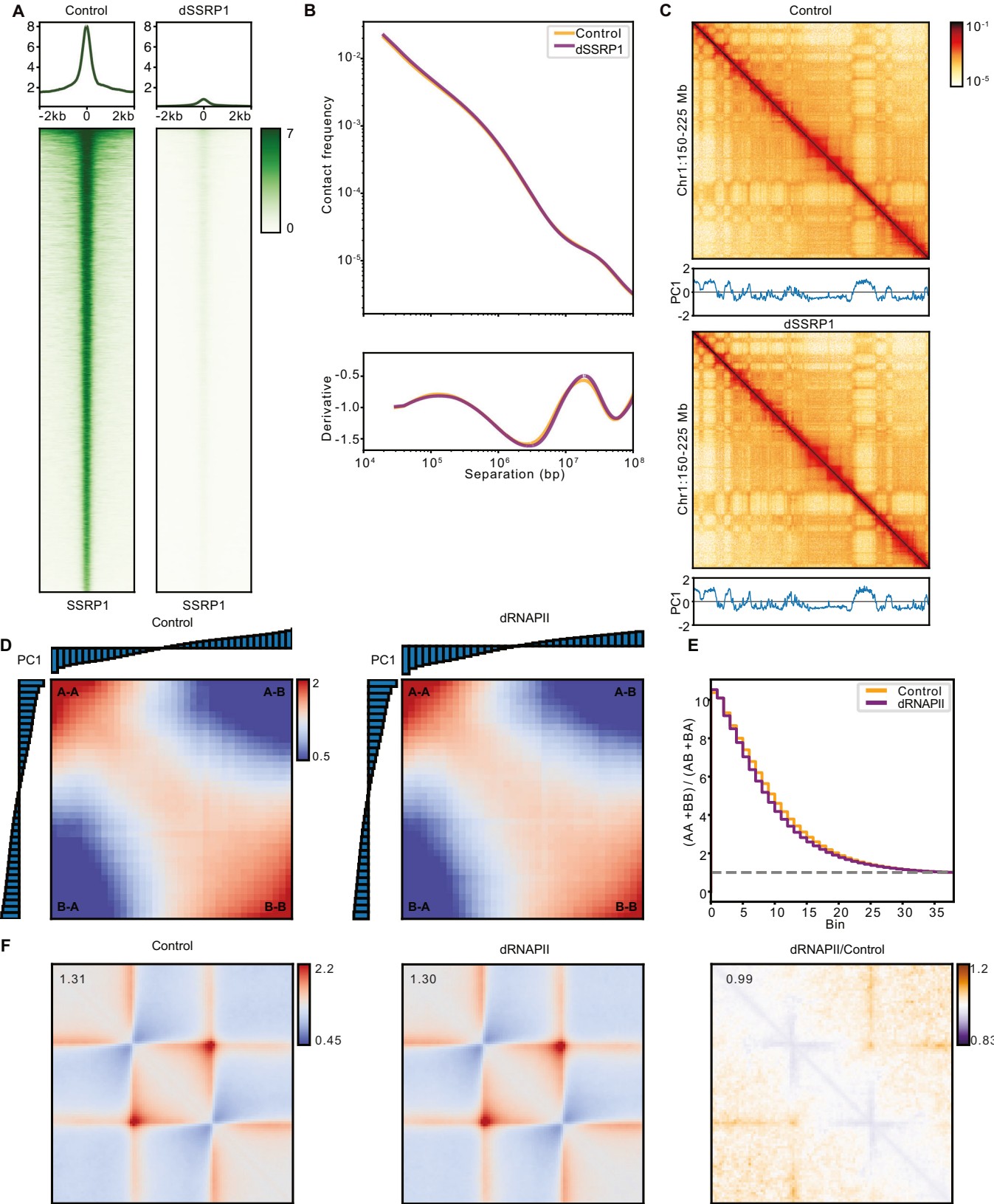

◄  **Figure EV1.  Rapid depletion of FACT leads to changes in 3D genome organization that are not secondary to changes in transcription.**

(A) SSRP1 ChIPmentation signal in control (left) and SSRP1-depleted (right) K562 cells. ChIPmentation data show the average of 2 biological replicates. (B) Micro-C contact frequency decay as a function of genomic distance (top) and its first derivative (bottom) in control (orange) and SSRP1-depleted (purple) cells. Data as in Fig. 1A. (C) Micro-C contact matrices (100 kb resolution) of an exemplary region on chromosome 1 in control (left) and SSRP1-depleted (right) cells. Below is the Eigenvector indicating A compartment ( > 0) or B compartment ( < 0). Data as in Fig. 1A. (D) Saddleplots displaying observed/expected intra- and intercompartment interactions of genomic regions stratified into 39 bins by Eigenvector decomposition in control (left) and RNAPII-depleted cells (right). Histograms show Eigenvalues of stratified bins. Micro-C data show the average of 2 biological replicates. (E) Step-plot showing the saddle strength profile (intracompartmental/intercompartmental interactions) in control (orange) and RNAPII-depleted (purple) cells. Data as in (D). (F) Mean observed/expected contact frequencies at TADs ($n = 2831$) in control (left) and RNAPII-depleted (middle) cells and their relative differences (right; purple indicates enriched contacts in control cells; orange indicates enriched contacts in RNAPII-depleted cells). TADs are rescaled to the same size. Average signal is reported in the top left. Data as in (D).

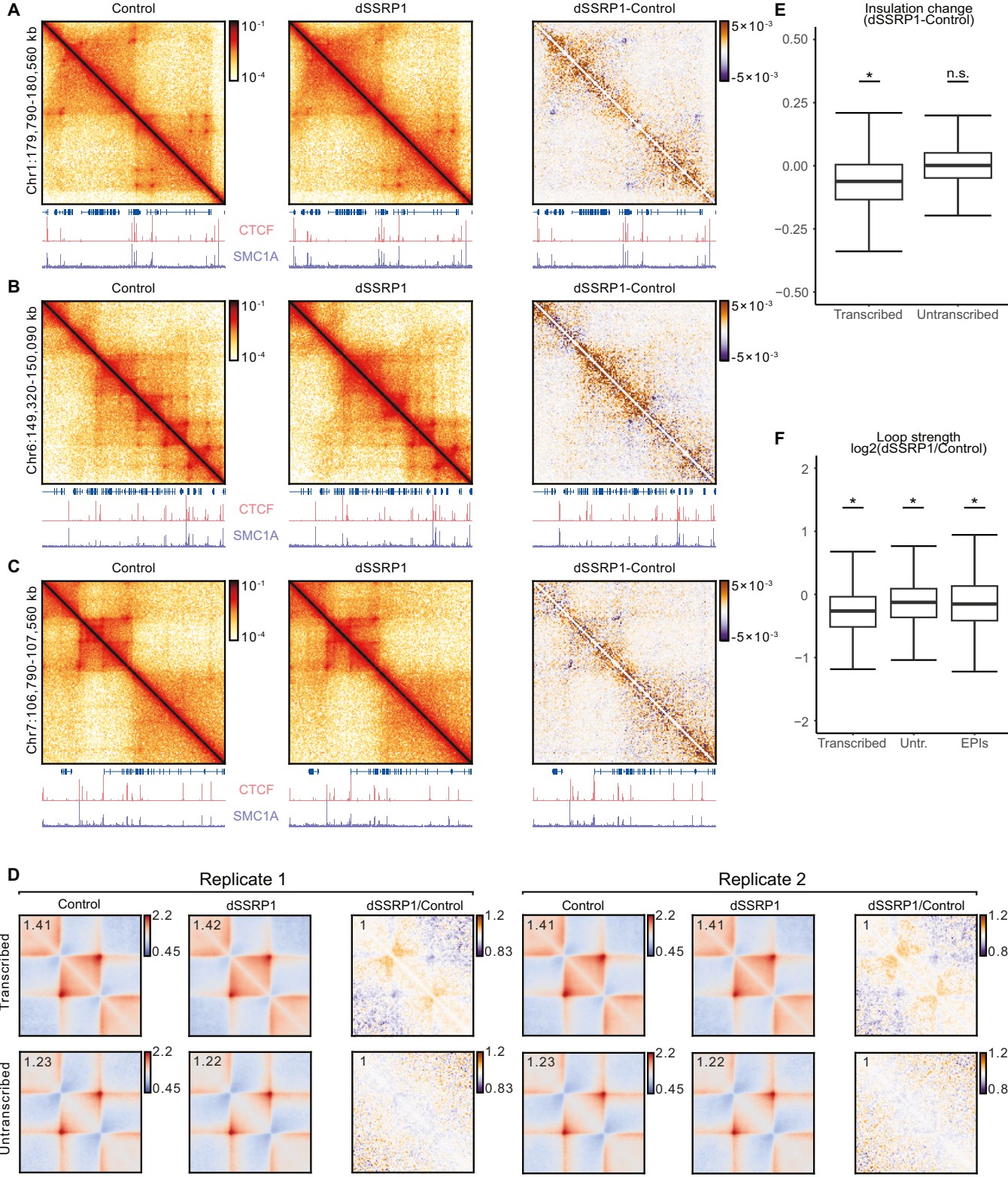

◀ **Figure EV2. Rapid depletion of FACT leads to significant changes in 3D genome organization that are consistent across genomic regions and replicates.**

(A) Micro-C contact matrices (5 kb resolution) of an exemplary region on chromosome 1 in control (left) and SSRP1-depleted (middle) K562 cells and their absolute differences (right; purple indicates enriched contacts in control cells; orange indicates enriched contacts in SSRP1-depleted cells). Gene annotation and ChIPmentation tracks for CTCF and SMC1A in control (left and right) and SSRP1-depleted cells (middle) are shown at the bottom. Data as in Fig. 2A. (B) Same as (A) for an exemplary region on chromosome 6. (C) Same as (A) for an exemplary region on chromosome 7. (D) Mean observed/expected contact frequencies at transcribed (top) and untranscribed (bottom) TADs in independent replicates of control (left) and SSRP1-depleted (middle) cells and their relative differences (right; purple indicates enriched contacts in control cells; orange indicates enriched contacts in SSRP1-depleted cells). TADs are rescaled to the same size. Average signal is reported in the top left. Data as in Fig. 2A. (E) Relative differences in insulation scores at transcribed (left; two-sided Wilcoxon rank-sum test; $n = 1811$, $P < 2.2e\text{-}16$) and untranscribed (right; two-sided Wilcoxon rank-sum test; $n = 1948$, $P < 2.2e\text{-}16$) TAD boundaries between control and dSSRP1-depleted cells. Boxplots show the interquartile range (IQR) and median of the data; whiskers indicate the minima and maxima within 1.5 * IQR; asterisks indicate statistical significance ($P < 0.05$); n.s. = non-significant. Data as in Fig. 2A. (F) Same as (E) for CTCF loop strength at transcribed (left; two-sided Wilcoxon rank-sum test; $n = 657$, $P < 2.2e\text{-}16$) and untranscribed (Untr.; middle; two-sided Wilcoxon rank-sum test; $n = 900$, $P < 2.2e\text{-}16$) TAD boundaries and enhancer–promoter interactions (right; two-sided Wilcoxon rank-sum test; $n = 423$, $P < 9.718e\text{-}11$).

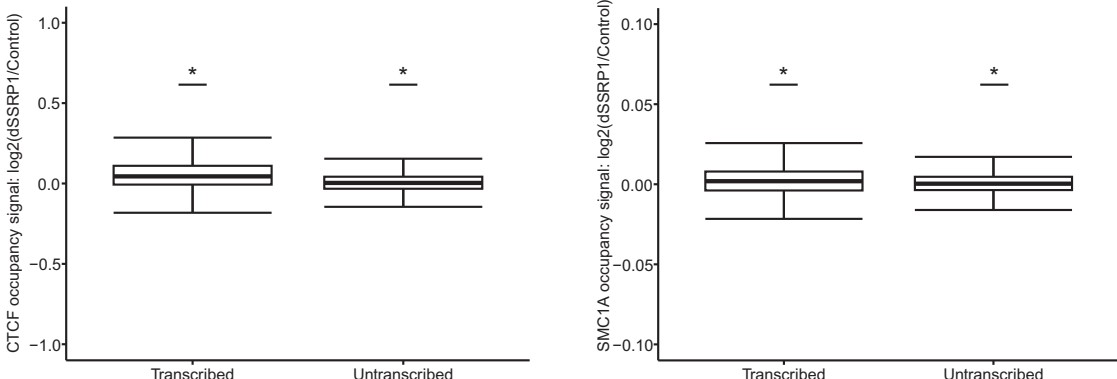

**Figure EV3.  The effects of FACT depletion on TAD organization are not dependent on reduced CTCF and cohesin occupancy.**

Relative differences in CTCF (left panel) and SMC1A (right panel) occupancy between control and dSSRP1-depleted K562 cells in transcribed (left box; two-sided Wilcoxon rank-sum test; $n = 22,681$, $P < 2.2e$-16) and untranscribed (right box; two-sided Wilcoxon rank-sum test; $n = 20,099$, $P < 2.2e$-16) regions. Boxplots show the interquartile range (IQR) and median of the data; whiskers indicate the minima and maxima within 1.5 * IQR; asterisks indicate statistical significance ($P < 0.05$). Data as in Fig. 3.

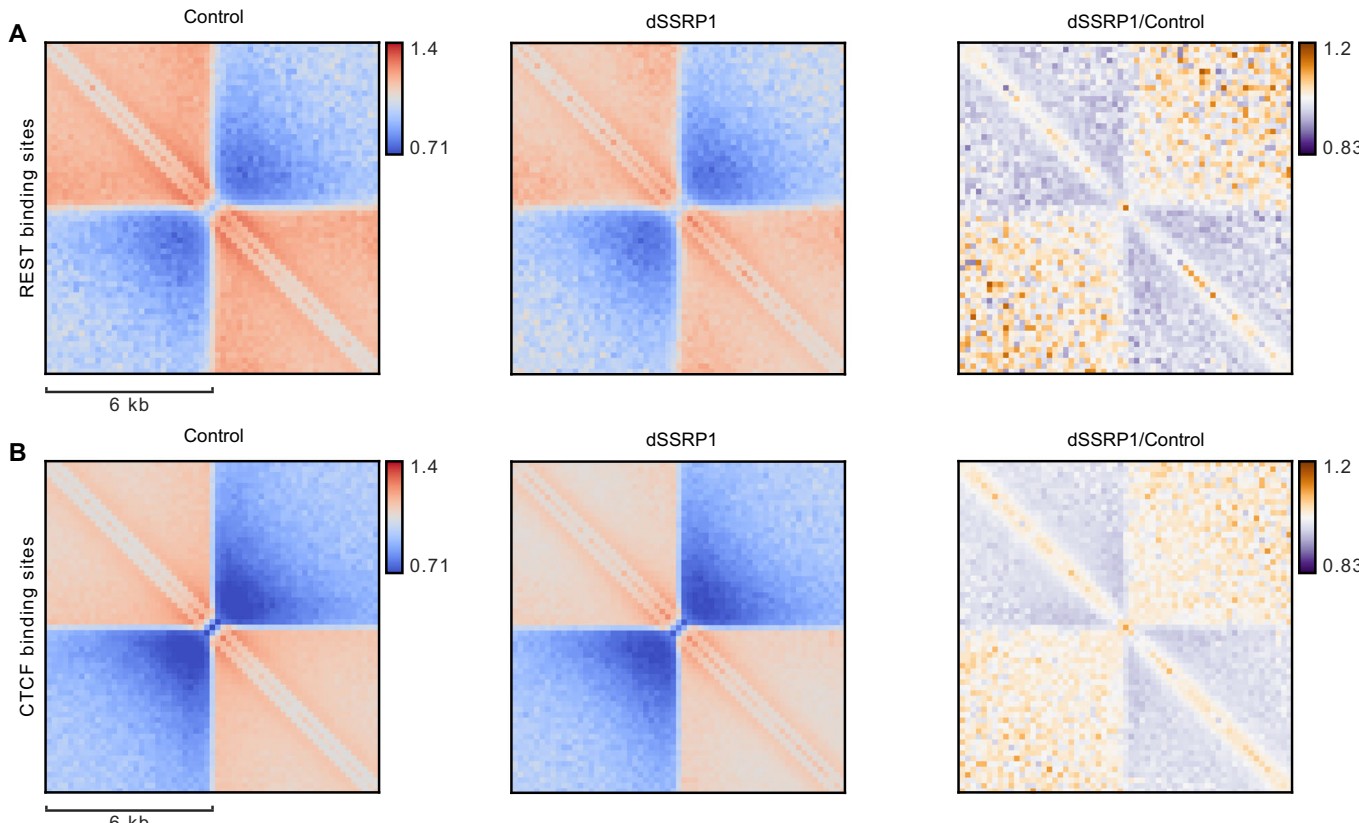

**Figure EV4.  Local insulation at REST and CTCF binding sites.**

(**A**) Mean observed/expected contact frequencies at REST binding sites (that do not overlap with CTCF binding sites; $n = 13,387$) in control (left) and SSRP1-depleted (middle) K562 cells and their relative differences (right; purple indicates enriched contacts in control cells; orange indicates enriched contacts in SSRP1-depleted cells). Note that insulation shown here extends over much shorter distances compared to insulation at CTCF-bound TAD borders shown in Fig. 1C. Data as in Fig. 2A. (**B**) Same as (**A**) at all CTCF binding sites across the genome (including CTCF binding sites that do not overlap with TAD borders; $n = 57,154$).

**Table EV1.  Micro-C mapping, deduplication, and contact statistics from 2 merged biological replicates.**

|  | Control | | dSSRP1 | |
| --- | --- | --- | --- | --- |
|  | **Read number** | **Percentage** | **Read number** | **Percentage** |
| Total read pairs | 2,033,336,706 | 100 | 1,830,972,576 | 100 |
| Mapped read pairs | 1,821,115,104 | 89.6 | 1,626,027,133 | 88.8 |
| Duplicate read pairs | 304,308,463 | 26.6 | 260,134,033 | 25.8 |
| Valid *trans* read pairs | 152,853,931 | 13.4 | 127,161,826 | 12.6 |
| Valid *cis* read pairs | 686,790,761 | 60 | 620,232,513 | 61.6 |

Percentages in top two rows are relative to total number of read pairs; percentages in bottom three rows are relative to total number of uniquely aligned read pairs.

                      