## [Peer Review File · Molecular Systems Biology]

FACT depletion demonstrates a role for nucleosome organization in TAD formation

Clemens Mauksch, Yi Zhu, Taras Velychko, Spyridoula Sagropoulou, Abrar Aljahani, Shyam Ramasamy, Kristina Zumer, and Marieke Oudelaar

Corresponding author(s): Marieke Oudelaar (marieke.oudelaar@mpinat.mpg.de)

Review Timeline:

Submission Date:	7th Jun 25
Editorial Decision:	21st Aug 25
Revision Received:	1st Oct 25
Accepted:	6th Oct 25

Editor: Poonam Bheda

Transaction Report:

This manuscript was transferred to Molecular Systems Biology following peer review at another journal.

We thank the Reviewers for their constructive and helpful feedback on our work. As discussed below, we have addressed all raised concerns, which we think has significantly improved the manuscript. Although not directly requested by the Reviewers, we have changed all differential analyses from Control/dSSRP1 to dSSRP1/Control, since we realized that this is more intuitive for most readers.

Reviewer # 1:

Nucleosome is the basic structural unit that organizes eukaryotic chromatin. I suggest a revision of this manuscript. Here are several points that I suggest to be clarified or explained:

Following the Reviewer's suggestions, we have incorporated extensive statistical analyses and re-analyzed the ChIP data. The additional analyses further support the findings described in the manuscript. We thank the Reviewer for these suggestions, which have helped us to strengthen the support for our conclusions.

1. There are many vague words such as "minor" "subtle" "not strongly" to describe the changes after the depletion of FACT. Are they "small yet statistically significant" or "insignificant" changes? Since these descriptions are quite confusing, I suggest adding some statistical data to enhance the claims. And explain why we should pay attention to some subtle changes but not the "minor" or "not strongly" changes.

We thank the Reviewer for raising this point. We have incorporated extensive statistical analyses, which support all reported observations and conclusions. More specifically, we have made the following changes to the manuscript:

- We show that FACT depletion significantly reduces the insulation scores of TAD boundaries and significantly reduces the contact frequencies of CTCF loops. This confirms our claims about the role of FACT in TAD insulation and CTCF loop formation. We have added these analyses to **Figure 4** and **Figure S2** and re-written the corresponding text accordingly.
- We show that FACT depletion significantly increases nucleosome fuzziness and decreases nucleosome occupancy, indicating a decrease in regular nucleosome organization at CTCF binding sites in absence of FACT. We have added these analyses to **Figure 4** and re-written the corresponding text accordingly.
- We show that FACT depletion does *not* lead to a significant overall reduction of CTCF and cohesin occupancy. This confirms that the changes in TAD insulation and CTCF loops are not due to reduced CTCF and cohesin occupancy. We have added these analyses to **Figure S3** and re-written the corresponding text accordingly. In line with these analyses, we observe the same effect on TAD insulation and CTCF looping in metaplots in which we only include TADs with boundaries at which CTCF/cohesin occupancy levels do not significantly change. This is shown in **Figure 3**. (Of note: we actually find a small but significant overall increase in CTCF and cohesin occupancy after FACT depletion. This is likely due to increased accessibility for protein binding due to loss of nucleosomes.)

2. The CTCF and cohesion occupancy were considered as "not strongly affected". Please show the comparison like the "TADs" data as the control/Dssrp1. In addition, tell us more definitively whether they were changed or not, because it will obviously affect the conclusion.

We have included a differential analysis in **Figure 3**. In line with the changes that we made to the other figures, we show this as dSSRP1/Control. As discussed above, we do not see a significant overall decrease in CTCF and cohesin occupancy. Furthermore, we show in **Figure 3** that the changes in protein occupancy cannot explain the changes in TAD insulation and CTCF looping, as we observe the same effects on TAD structure in metaplots for which only TADs with stable (i.e., not significantly changed) CTCF and cohesin occupancy are included. We have clarified this in the text. The conclusions of the manuscript therefore remain unchanged.

3. The CTCF binding sites in transcribed regions were the data from the "ENCODE" which represented the statistic sites for CTCF binding. It is unclear whether CTCF still binds to these sites in the FACT-depleted cells. Please use the "CHIPmentation" data to show the CTCF binding (which could also include the impact of CTCF binding on the nucleosome positioning).

The Reviewer is correct that we had used the ENCODE CTCF ChIP-seq data to align the Micro-C and MNase-seq data in the previous version of the manuscript. We understand the Reviewer's concern with respect to this strategy. We have therefore re-analyzed the Micro-C and MNase-seq data. We now use our CHIPmentation data as the basis for these analyses to make sure that we are only looking at sites that are bound in our cell line in control and SSRP1-depleted conditions. To ensure that we are using robust/valid peaks only, we have intersected our CHIPmentation peaks with the ENCODE peaks and only retained CHIPmentation peaks that are also present in ENCODE peaks (which is true for the vast majority of peaks). We have used the ENCODE data (which are very deeply sequenced and more precise) to determine the exact center location of the CHIPmentation peaks and used these coordinates for aligning the Micro-C and MNase-seq data. The new analysis workflow does not change our conclusions.

Reviewer #2:

In this manuscript, Mauksch and colleagues combine rapid depletion of the FACT complex (SSRP1-dTAG degron) with genome-wide assays to determine its impact on chromatin structure and organization. The study reveals a minor increase in separation between chromatin compartments and a comparably minor reduction in TAD insulation. The latter coincides with some reduction in nucleosome positioning signal, particularly at CTCF binding sites in transcribed regions. This study follows up on a previous study (Zumer et al, Mol Cell 2024), where an investigation of chromatin reorganization in the same cells was performed but limited to a few chromosomal domains.

We thank the Reviewer for their critical and thorough assessment of our work, which has helped us to clarify the main findings of the paper and the advance compared to previous work. As discussed in more detail below, we have added additional analyses to demonstrate that the effects of FACT depletion on TAD structure are robust and reproducible, and to clarify the cause-consequence relationship between nucleosome organization and TAD structure. In addition, we have clarified the value of this manuscript in relation to previous work.

Having read the manuscript, I think it's not suited for publication in this journal for three reasons:

1. The reported genome-wide differences are very minor: around 10% reduction in key-panels 1C and 2B, a similar very minor reduction in figure 4 and a nearly indistinguishable pattern

in panel 2A. These findings do not support the at times strong conclusions (Lines 128-143: Comparison of transcribed and untranscribed TADs clearly shows that changes in TAD structure predominantly occur in transcribed regions. Lines 144-145: Accordingly, we observe a clear loss in CTCF loops in transcribed regions only. Lines 177-178: FACT depletion perturbs nucleosome positioning at CTCF binding sites in transcribed regions). Rather, I'm not convinced that these mild differences could not be explained by mere variation between experiments or secondary effects of the FACT-depletion system.

We thank the Reviewer for raising this point. After submission of our manuscript, we realized that the annotation that we used for our analysis is not optimal. In the previous version of our manuscript, we had defined "untranscribed regions" in a very strict way, and considered everything else "transcribed", which means that the category of "transcribed regions" contained a lot of regions that were not actually transcribed but only contained low-level transcriptional noise. We have now corrected this. Importantly, in our new analysis, the effect size is doubled (**Rebuttal Figure 1**).

Rebuttal Figure 1. TAD structure in control and FACT-depleted (dSSRP1) conditions in "transcribed regions" according to (A) the new annotation and (B) the old annotation, as described above.

We would like to stress that this is really not a minor / irrelevant effect. To put this into context, we copied a figure from Iurlaro et al. (*Nature Genetics* 2024) below (**Rebuttal Figure 2**). This paper focuses on the role of Bptf in 3D genome organization, which the authors contrast to previous work on Snf2h (Barisic et al. *Nature* 2019) and CTCF (Nora et al. *Cell* 2017). The effect of FACT depletion is larger than the effect of Bptf deletion and similar to Snf2h deletion.

Rebuttal Figure 2. The effects of *Bptf*, *Snf2h*, and CTCF perturbation on TAD formation. Figure copied from Iurlaro et al. *Nature Genetics* 2024.

We agree with the Reviewer that the region shown in **Figure 2A** is not the best example to illustrate the changes in TAD structure in transcribed regions. We have therefore replaced this figure with a different region (pasted below) and added additional example loci to **Figure S2A-C** to illustrate the changes in TAD insulation and CTCF looping upon FACT depletion.

Main Figure 2: FACT depletion weakens TAD insulation and CTCF loops in transcribed regions. (A) Micro-C contact matrices (5 kb resolution) of an exemplary region on chromosome 11 in control (left) and SSRP1-depleted (middle) cells and their absolute differences (right; purple indicates enriched contacts in control cells; orange indicates enriched contacts in SSRP1-depleted cells). Dashed highlights indicate regions in which loop strength is reduced. Gene annotation and ChIPmentation tracks for CTCF and SMC1A in control (left and right) and SSRP1-depleted cells (middle) are shown at the bottom.

Finally, we have performed the Micro-C experiments in 2 independent replicates, which are extremely consistent. We can therefore rule out that our results can be explained by variation between experiments. We have included analyses of the individual replicates to clarify this to **Figure S2D** (pasted below).

Supplemental Figure S2: Rapid depletion of FACT leads to significant changes in 3D genome organization that are consistent across genomic regions and replicates. (D) Mean observed/expected contact frequencies at transcribed (top) and untranscribed (bottom) TADs in independent replicates of control (left) and SSRP1-depleted (middle) cells and their relative differences (right; purple indicates enriched contacts in control cells; orange indicates enriched contacts in SSRP1-depleted cells). TADs are rescaled to the same size. Average signal is reported in the top left.

As we describe in the next point in more detail, we have also performed additional analyses to show that the effects that we describe are not due to secondary effects of the FACT depletion system.

2. Despite the claim (lines 216-217) "changes in nucleosome positioning upon FACT depletion indicates that regular nucleosome positioning contributes to domain insulation", the observations remain correlative without insights into the mechanistic underpinnings or a formal confirmation of cause/consequence. This falls below the scrutiny expected for publication in this journal.

To investigate the cause-consequence relationship between nucleosome organization and TAD structure, we analyzed the relationship between these two features in more detail. Importantly, when we subset our data into quartiles ranging from the most highly transcribed regions to the lowest, we observe that the degree to which nucleosome organization and TAD structure/CTCF loops are perturbed is very consistent and commensurate (**Figure 4A-C**; pasted below), in line with a causal relationship. The finding that nucleosome organization contributes to TAD structure is important for our understanding of the molecular mechanisms that drive higher-order 3D genome organization.

Main Figure 4: FACT depletion leads to commensurate changes in nucleosome organization and TAD structure. (A) Metaplots showing the mean smoothed dyad signal (derived from MNase-seq data) at TAD boundaries bound by CTCF, categorized into quartiles based on transcriptional activity (0-25%: $n=1307$; 25-50%: $n=1306$; 50-75%: $n=1307$; 75-100%: $n=1307$), in control (orange) and SSRP1-depleted (purple) cells.

Profiles are centered on the midpoint of CTCF binding sites and aligned based on the orientation of the CTCF motifs. (B) Relative differences in mean observed/expected contact frequencies at TADs of which both boundaries are in the same quartile of transcriptional activity (0-25%: n=165; 25-50%: n=85; 50-75%: n=84; 75-100%: n=211) between control and SSRP1-depleted cells (purple indicates enriched contacts in control cells; orange indicates enriched contacts in SSRP1-depleted cells). TADs are rescaled to the same size. Average signal is reported in the top left. (C) Relative differences in mean observed/expected contact frequencies at CTCF loops of which both anchors are in the same quartile of transcriptional activity (0-25%: n=232; 25-50%: n=226; 50-75%: n=251; 75-100%: n=371) between control and SSRP1-depleted cells (purple indicates enriched contacts in control cells; orange indicates enriched contacts in SSRP1-depleted cells).

3. Despite the genome-wide nature of the studies, few new insights are added to the preceding study (Zumer et al, Mol Cell 2024). Compare for instance figure 4 vs panel 1I in Zumer et al. or panel 2A vs panel 2D in Zumer. The incremental nature of the conclusions falls below the expected standards for publication in this journal.

We thank the Reviewer for the opportunity to clarify this point. The paper by Zumer et al. focusses on the role of FACT in transcription. The 3D genome experiments in this paper are targeted and our analysis is focused on small-scale features of genome organization. With the strategy used in Zumer et al., we were not able to detect any changes in higher-order 3D genome organization, which was also not the focus of this work. We discuss this in the introduction of our manuscript.

Here, we leverage the system that we developed in Zumer et al. to study the relationship between nucleosome organization and TAD formation. Using high-resolution, genome-wide analysis, we discover that the organization of nucleosomes influences TAD formation. This finding is conceptually very novel and important for the field, as nucleosome organization is currently not considered as a driver of higher-order genome folding in mammals.

Reviewer #3:

The manuscript by Mauksch et al describes the effect of acute depletion of FACT on 3D genome organization using HiC. The main claims are: 1) depletion of FACT leads to weakening of TAD insulation in transcribed regions without affecting CTCF binding; 2) This effect is consequent to the role of FACT in nucleosome positioning over transcribed regions. The manuscript is very well written, clear, concise and easy to read. However, I am not convinced that the data support the claims.

Following the Reviewer's suggestions, we have incorporated additional quantitative and statistical analyses and re-written the text to describe the changes in nucleosome organization more accurately. The additional analyses further support the findings described in the manuscript. We thank the Reviewer for these suggestions, which have helped us to strengthen the support for our conclusions and clarify our manuscript.

1- Unlike the effect of FACT depletion on the strength of TAD boundaries, which –despite being very subtle– is nevertheless visible on the differential maps (control/ depletion; Figures 1C and 2), the effect of FACT depletion on nucleosome positioning, however, is not convincing (Figure 4). Figure 4 shows a very small difference in amplitude for the depletion condition, which may indicate a minor effect on nucleosome occupancy, but I am not able to see any convincing difference in positioning. If phasing and /or fuzziness is affected, it is not apparent in these plots. Can the authors do more analyses (ideally quantitative and including statistical

assessments) to convince me that these parameters are affected by FACT depletion? I also do not see any difference between transcribed and non-transcribed region in Figure 4.

We thank the Reviewer for raising this point. As discussed in more detail in response to the Reviewer's third comment, we had used the term "nucleosome positioning" in the previous version of our manuscript in a broad sense. We realize now that this is very confusing. We intended to state that FACT depletion leads to changes in the organization of nucleosomes around CTCF binding sites. The Reviewer is completely correct that there are no clear changes in nucleosome positioning (according to a strict definition of nucleosome positioning, i.e. there is no position shift) and has noticed correctly that there are changes in nucleosome occupancy and fuzziness.

To clarify where the biggest changes in nucleosome organization occur and to relate the changes in nucleosome organization directly to changes in TAD structure, we have segregated our data into quartiles based on transcriptional activity. This analysis reveals that the largest changes in nucleosome occupancy and fuzziness occur in highly transcribed regions, where we also observe the largest changes in TAD insulation and CTCF looping. In the quartiles with lower transcriptional activity, we see smaller changes in these features.

Following the Reviewer's helpful suggestion, we have performed quantitative and statistical analyses. This shows that the changes in nucleosome occupancy and fuzziness, as well as the changes in TAD insulation and CTCF loops, are highly significant. Somewhat surprisingly, we find that these changes are significant across all quartiles (with the exception of insulation). However, it is very clear, both from the metaplots and from the quantifications, that the changes are biggest in the transcribed regions. Importantly, we also observe that the changes in nucleosome organization and TAD structure are commensurate across the quartiles, indicative of a direct relationship between nucleosome organization and TAD structure. We have included these new analyses in **Figure 4** (pasted below) and have re-written the corresponding text accordingly.

Main Figure 4: FACT depletion leads to commensurate changes in nucleosome organization and TAD structure. (A) Metaplots showing the mean smoothed dyad signal (derived from MNase-seq data) at TAD boundaries bound by CTCF, categorized into quartiles based on transcriptional activity (0-25%: $n=1307$; 25-50%: $n=1306$; 50-75%: $n=1307$; 75-100%: $n=1307$), in control (orange) and SSRP1-depleted (purple) cells. Profiles are centered on the midpoint of CTCF binding sites and aligned based on the orientation of the CTCF motifs. (B) Relative differences in mean observed/expected contact frequencies at TADs of which both boundaries are in the same quartile of transcriptional activity (0-25%: $n=165$; 25-50%: $n=85$; 50-75%: $n=84$; 75-100%: $n=211$) between control and SSRP1-depleted cells (purple indicates enriched contacts in control cells; orange indicates enriched contacts in SSRP1-depleted cells). TADs are rescaled to the same size. Average signal is reported in the top left. (C) Relative differences in mean observed/expected contact frequencies at CTCF loops of which both anchors are in the same quartile of transcriptional activity (0-25%: $n=232$; 25-50%: $n=226$; 50-75%: $n=251$; 75-100%: $n=371$) between control and SSRP1-depleted cells (purple indicates enriched contacts in control cells; orange indicates enriched contacts in SSRP1-depleted cells). (D) Relative differences in nucleosome fuzziness at TAD boundaries bound by CTCF, categorized into quartiles based on transcriptional activity (0-25%: $n=14174$; 25-50%: $n=14863$; 50-75%: $n=15871$; 75-100%: $n=16050$), between control and dSSRP1-depleted cells. Boxplots show the interquartile range (IQR) and median of the data; whiskers indicate the minima and maxima within $1.5 * IQR$; asterisks indicate statistical significance ($p < 0.05$); n.s. = non-significant. (E) Same as panel D for nucleosome occupancy. (F) Same as panel D for insulation (0-25%: $n=1307$; 25-50%: $n=1306$; 50-75%: $n=1307$; 75-100%: $n=1307$). Note that the difference is not log-transformed as the insulation

score represents a log-transformed value. (G) Same as panel D for loop strength (0-25%: n=232; 25-50%: n=226; 50-75%: n=251; 75-100%: n=371).

2- I would also like to see relative difference heat maps and metagenes for CTCF occupancy (Figure 3). Since the effect on TADs required relative difference maps to be visible, it would be fair to treat the CTCF occupancy data the same way before dismissing any effect of FACT depletion on this parameter. There is not strong effect for sure, but for as far as I can tell the effect on TADs and nucleosomes are very subtle as well.

We have included differential analyses, quantifications, and statistical analyses of CTCF and cohesin occupancy in **Figure 3** and **Figure S3**. These analyses clearly show that FACT depletion does not lead to a significant overall reduction of CTCF and cohesin occupancy and therefore confirm that the changes in TAD insulation and CTCF loops are not due to reduced CTCF and cohesin occupancy. In line with these analyses, we observe the same effect on TAD insulation and CTCF looping in metaplots in which we only include TADs with boundaries at which CTCF/cohesin occupancy levels do not significantly change. This is also shown in **Figure 3**. (Of note: we actually find a small but significant overall increase in CTCF and cohesin occupancy after FACT depletion. This is likely due to increased accessibility for protein binding due to loss of nucleosomes.) Together, these analyses confirm that we cannot explain the changes in TAD insulation and CTCF looping by reduced CTCF and/or cohesin occupancy.

3- Finally, I am not comfortable with the claim that the effect on TADs stems from a positioning defect. Notwithstanding the apparent lack of effect on positioning mentioned above, the logic to attribute the effect on TADs to nucleosome positioning is flawed. Indeed, because FACT depletion leads to so diverse chromatin, transcriptional and replication defects, one simply can not draw a causal link here. The authors have a good argument to refute transcription as the explanation, but other possibilities remain. Among others, I would argue that since the MNase data reveals a much more obvious occupancy defect than a positioning defect, one should –if anything– consider (if not favor) defect in occupancy as a possibility. The authors raise this possibility in the Discussion but yet, the title of the manuscript claims that changes in positioning as causal.

We are very grateful to the Reviewer for raising this point and we apologize for the confusion! We intended to use the term "nucleosome positioning" in a broad sense and to refer with this term to all changes in nucleosome organization that we observe. We fully agree with the Reviewer that this is very confusing. As discussed in response to the Reviewer's first comment, we indeed do not observe clear changes in nucleosome positioning in a strict sense. We do however observe significant changes in nucleosome fuzziness and occupancy. We have rewritten the text (incl. title) to clarify this. Instead of "nucleosome positioning", we now use the term "nucleosome organization" to refer to the observed changes in nucleosome arrangements in general and use the correct terms when describing the more specific defects.

21st Aug 2025

Manuscript Number: MSB-2025-13164

Title: FACT depletion demonstrates a role for nucleosome organization in TAD formation

Dear Dr. Oudelaar,

Thank you for the submission of your revised manuscript to Molecular Systems Biology. We have now received the enclosed reports from the referees that were asked to assess it. As you will see the reviewers are now globally supportive and I am pleased to inform you that we will be able to accept your manuscript pending the following final amendments:

- 1) Please download the EMBO Press "Author Checklist" and complete all relevant questions. This file should be uploaded with your submission. This file can be downloaded from our website at:
<https://www.embopress.org/page/journal/17444292/authorguide>
- 2) In the main manuscript file, please rename "Summary" to "Abstract".
- 3) Please include keywords to max. 5.
- 4) Please remove the Resource Availability section except for the Data Availability statement (the Lead Contact and Materials Availability sections are not needed). Please format the Data availability section according to the example below. The statements that the study does not report original code and that information required to reanalyze the data are not necessary: "The datasets and computer code produced in this study are available in the following databases:
 - Chip-Seq data: Gene Expression Omnibus GSE46748 (<https://www.ncbi.nlm.nih.gov/geo/query/acc.cgi?acc=GSE46748>)
 - Modeling computer scripts: GitHub (<https://github.com/SysBioChalmers/GECKO/releases/tag/v1.0>)
 - [data type]: [full name of the resource] [accession number/identifier] ([doi or URL or identifiers.org/DATABASE:ACCESSION])"
- 5) Please release the dataset GSE284561 so that it is now publicly available.
- 6) Please rename "Declaration of Interests" to "Disclosure and competing interests statement". We updated our journal's competing interests policy in January 2022 and request authors to consider both actual and perceived competing interests. Please review the policy <https://www.embopress.org/competing-interests> and update your competing interests if necessary.
- 7) Please remove the author contributions from the manuscript and specify author contributions in our submission system. CRedit has replaced the traditional author contributions section because it offers a systematic machine-readable author contributions format that allows for more effective research assessment. You are encouraged to use the free text boxes beneath each contributing author's name to add specific details on the author's contribution. More information is available in our guide to authors:
<https://www.embopress.org/page/journal/17574684/authorguide#authorshippinguidelines>
- 8) References: Please correct the reference citation in the reference list to be alphabetical (not numerical). Where there are more than 10 authors on a paper, only the first 10 should be listed, followed by "et al.". Please check "Author Guidelines" for more information.
<https://www.embopress.org/page/journal/17574684/authorguide#referencesformat>
- 9) Our journal encourages inclusion of *data citations in the reference list* to directly cite datasets that were re-used and obtained from public databases. Data citations in the article text are distinct from normal bibliographical citations and should directly link to the database records from which the data can be accessed. In the main text, data citations are formatted as follows: "Data ref: Smith et al, 2001" or "Data ref: NCBI Sequence Read Archive PRJNA342805, 2017". In the Reference list, data citations must be labeled with "[DATASET]". A data reference must provide the database name, accession number/identifiers and a resolvable link to the landing page from which the data can be accessed at the end of the reference. Further instructions are available at .
- 10) In the Methods, please take care of the following:
 - The "Star Methods" section should be renamed to "Methods".
 - Please also be sure to include a sentence in the Methods as to whether or not the cell lines used were recently authenticated.
 - Please ensure that a statement on whether or not blinding was done is included in the Methods even if no blinding was done. Please also be sure to include this information in the Author Checklist and where it can be found in the manuscript.
- 11) When submitting your revised manuscript, please do not include the Reagents and Tools Table in the Methods section of the manuscript but upload it as a separate file choosing the file type "Reagent Table". Please be sure to use our template, which you can download from our author guidelines:
<https://www.embopress.org/page/journal/14693178/authorguide#structuredmethods>.
- 12) Please place individual sections of the manuscript in the following order: Title page - Abstract & Keywords - Introduction - Results - Discussion - Methods - Data Availability - Acknowledgements - Disclosure and Competing Interests Statement - References - Figure Legends - Expanded View Figure Legends.
- 13) For the figures and figure legends, please take care of the following:
 - You have Figures S1-S4 which can be made into Expanded View figures. In this case, each figure will still need to fit onto one page and be renamed as Figure EV1, etc. The legends should stay in the manuscript, with the heading Expanded View Figures Legends, and placed after the main figure legends. Alternatively, you may compile these into a single Appendix PDF and renamed to Appendix Figure S1-S4 with legends below the appropriate figure. In either case, the callouts to the figure should be

updated, as well as the titles of the files in our submission system.

- Please remove all figures from main manuscript file and leave only main and EV figure legends placed after the references. EV figure legends should be placed in their own section with the heading Expanded View Figures Legends, and placed after the main figure legends. Main figures and EV figures should be uploaded as individual, high-resolution files. Please check "Author Guidelines" for more information: <https://www.embopress.org/page/journal/17574684/authorguide#figureformat>

- All figures and figure panels should be called out sequentially.

- Please define the annotated p values ****/**/*/* as well as provide the exact p-values for the same in the legend of figure 4D-G; S2 E, F; S3 as appropriate.

- Please indicate the statistical test used for data analysis in the legends of figures 4D-G; S2 E, F; S3

14) Supplementary Table 1 should be renamed to Table 1 (if it's editable) and placed below main and EV figure legends.

Alternatively it can be uploaded as Table EV1 or included in the Appendix PDF as Appendix Table S1. The callout should be updated as appropriate callout; in either of the latter two cases, the table legend should be above the table.

15) Please ensure that all funding sources are entered into the manuscript submission system. Currently the following are missing in our submission system: the Max Planck Society; European Research Council (Starter Grant 3D-REG 101115401); via SFB 1565 (project 469281184/P02); the PhD program "Genome Science" - International Max Planck Research School at the Georg August University Göttingen; 548 the MSc/PhD program "Molecular Biology" - International Max Planck Research School at the Georg August University Göttingen

16) Synopsis:

- Synopsis image: Please provide a graphic that summarises the main findings of the manuscript on a glance and upload it as a high-resolution jpeg file 550 pixels wide x (300-600) pixels high.

- Synopsis text: Please provide a separate word document including a short standfirst (maximum of 300 characters, including spaces) and up to 5 bullet points to summarise the key NEW findings. They should be designed to be complementary to the abstract - i.e. not repeat the same text. We encourage inclusion of key acronyms and quantitative information (maximum of 30 words / bullet point). Please use the passive voice.

17) Source Data: Please ensure that a completed Source Data checklist is uploaded as a Related Manuscript File Instructions for Source Data and the Source Data checklist will be sent to you in a separate email shortly. Source Data should be organized as a single source data file (zipped) per figure for main figures (all EV and/or Appendix figure Source Data can be included in a single folder), with the panels clearly visible in the folder structure instead of a single excel file for all Source Data. e.g. all the Source data files for figure 1 need to be saved in a single folder and this needs to be zipped and then uploaded as "SD figure 1.zip" file.

18) As part of the EMBO Publications transparent editorial process initiative (see our policy here:

https://www.embopress.org/transparent-process#Review_Process), Molecular Systems Biology will publish online a Peer Review File (PRF) to accompany accepted manuscripts. This file will be published in conjunction with your paper and will include the anonymous referee reports, your point-by-point response and all pertinent correspondence relating to the manuscript. Let us know whether you agree with the publication of the PRF and as here, if you want to remove or not any figures from it prior to publication. Please note that the Authors checklist will be published at the end of the PRF.

19) After your paper is published, we may promote it on social media. If you have any handles or hashtags for Bluesky you would like included, please let us know.

20) Please provide a point-by-point letter INCLUDING my comments and your detailed responses (as Word file).

I look forward to reading a new revised version of your manuscript as soon as possible.

Yours sincerely,

Poonam Bheda, PhD
Scientific Editor
Molecular Systems Biology

Reviewer #1:

Based on the revised manuscript and the accompanying point-by-point response, I find that the authors have addressed the reviewers' concerns in a generally thorough and satisfactory manner. They have provided direct experimental evidence with appropriate statistical support, which effectively clarifies the previously ambiguous claims regarding TAD insulation and nucleosome organization. Furthermore, the addition of a transcriptional quartile-based analysis lends a more structured rationale to the observed correlation between nucleosome perturbation and changes in chromatin architecture.

Although the conclusions remain correlative rather than mechanistically definitive, the study offers conceptual novelty by delineating the role of nucleosome organization independently of CTCF/cohesin binding. Overall, the revisions have substantially improved the manuscript, bringing it to a standard appropriate for publication in Molecular Systems Biology.

Reviewer #2:

The authors have provided a point-by-point response to reviewers that answers and clarifies most of their concerns.

The main criticisms of reviewers included:

- 1) If the small effects in genome topology (TAD boundary insulation and loop formation) detected upon FACT degradation were statistically significant and reproducible. The authors have added statistics, additional analysis grouping transcribed regions into different quartiles and additional replicates to confirm their findings.
- 2) If degradation of FACT did or did not disrupt CTCF and Cohesin binding from chromatin. The authors have revisited their ChIPmentation data set and analysis and confirmed that this is not the case.
- 3) A confusion regarding nucleosome positioning versus occupancy and if indeed the effects on these parameters were significant upon FACT degradation. The authors have corrected the terms and verified that indeed positioning is not altered but occupancy is, even if the reduction is small. The authors changed the title to "nucleosome organization", I would suggest calling it occupancy as this is the more accurate term as nucleosome organization can mean many different things.

Editor

1) Please download the EMBO Press "Author Checklist" and complete all relevant questions. This file should be uploaded with your submission. This file can be downloaded from our website at: <https://www.embopress.org/page/journal/17444292/authorguide>

Done.

2) In the main manuscript file, please rename "Summary" to "Abstract".

Done.

3) Please include keywords to max. 5.

Done (3D genome organization, Nucleosome, Topologically associating domain (TAD), FACT, CTCF).

4) Please remove the Resource Availability section except for the Data Availability statement (the Lead Contact and Materials Availability sections are not needed). Please format the Data availability section according to the example below. The statements that the study does not report original code and that information required to reanalyze the data are not necessary:

"The datasets and computer code produced in this study are available in the following databases:
- Chip-Seq data: Gene Expression Omnibus GSE46748
(<https://www.ncbi.nlm.nih.gov/geo/query/acc.cgi?acc=GSE46748>)
- Modeling computer scripts: GitHub (<https://github.com/SysBioChalmers/GECKO/releases/tag/v1.0>)
- [data type]: [full name of the resource] [accession number/identifier] ([doi or URL or identifiers.org/DATABASE:ACCESSION])"

Done. Please note that we did not produce computer code and have therefore updated the statement to "The datasets produced in this study are available in the following databases...".

5) Please release the dataset GSE284561 so that it is now publicly available.

Done.

6) Please rename "Declaration of Interests" to "Disclosure and competing interests statement". We updated our journal's competing interests policy in January 2022 and request authors to consider both actual and perceived competing interests. Please review the policy <https://www.embopress.org/competing-interests> and update your competing interests if necessary.

Done.

7) Please remove the author contributions from the manuscript and specify author contributions in our submission system. CRediT has replaced the traditional author contributions section because it offers a systematic machine-readable author contributions format that allows for more effective research assessment. You are encouraged to use the free text boxes beneath each contributing author's name to add specific details on the author's contribution. More information is available in our guide to authors: <https://www.embopress.org/page/journal/17574684/authorguide#authorshipguidelines>

Done.

8) References: Please correct the reference citation in the reference list to be alphabetical (not numerical). Where there are more than 10 authors on a paper, only the first 10 should be listed, followed by "et al.". Please check "Author Guidelines" for more information.

<https://www.embopress.org/page/journal/17574684/authorguide#referencesformat>

Done.

9) Our journal encourages inclusion of *data citations in the reference list* to directly cite datasets that were re-used and obtained from public databases. Data citations in the article text are distinct from normal bibliographical citations and should directly link to the database records from which the data can be accessed. In the main text, data citations are formatted as follows: "Data ref: Smith et al, 2001" or "Data ref: NCBI Sequence Read Archive PRJNA342805, 2017". In the Reference list, data citations must be labeled with "[DATASET]". A data reference must provide the database name, accession number/identifiers and a resolvable link to the landing page from which the data can be accessed at the end of the reference. Further instructions are available at

<https://www.embopress.org/page/journal/17574684/authorguide#referencesformat>.

Done.

10) In the Methods, please take care of the following:

- The "Star Methods" section should be renamed to "Methods".

Done.

- Please also be sure to include a sentence in the Methods as to whether or not the cell lines used were recently authenticated.

Done.

- Please ensure that a statement on whether or not blinding was done is included in the Methods even if no blinding was done. Please also be sure to include this information in the Author Checklist and where it can be found in the manuscript.

Done.

11) When submitting your revised manuscript, please do not include the Reagents and Tools Table in the Methods section of the manuscript but upload it as a separate file choosing the file type "Reagent Table". Please be sure to use our template, which you can download from our author guidelines: <https://www.embopress.org/page/journal/14693178/authorguide#structuredmethods>.

Done.

12) Please place individual sections of the manuscript in the following order: Title page - Abstract & Keywords - Introduction - Results - Discussion - Methods - Data Availability - Acknowledgements -

Disclosure and Competing Interests Statement - References - Figure Legends - Expanded View Figure Legends.

Done.

13) For the figures and figure legends, please take care of the following:

- You have Figures S1-S4 which can be made into Expanded View figures. In this case, each figure will still need to fit onto one page and be renamed as Figure EV1, etc. The legends should stay in the manuscript, with the heading Expanded View Figures Legends, and placed after the main figure legends. Alternatively, you may compile these into a single Appendix PDF and renamed to Appendix Figure S1-S4 with legends below the appropriate figure. In either case, the callouts to the figure should be updated, as well as the titles of the files in our submission system.

Done.

- Please remove all figures from main manuscript file and leave only main and EV figure legends placed after the references. EV figure legends should be placed in their own section with the heading Expanded View Figures Legends, and placed after the main figure legends. Main figures and EV figures should be uploaded as individual, high-resolution files. Please check "Author Guidelines" for more information: <https://www.embopress.org/page/journal/17574684/authorguide#figureformat>

Done.

- All figures and figure panels should be called out sequentially.

Done.

- Please define the annotated p values ****/***/**/* as well as provide the exact p-values for the same in the legend of figure 4D-G; S2 E, F; S3 as appropriate.

Done. Please note that many p-values are $p < 2.2e-16$, which is the lower limit of the analysis software used, and are therefore indicated as such.

- Please indicate the statistical test used for data analysis in the legends of figures 4D-G; S2 E, F; S3

Done.

14) Supplementary Table 1 should be renamed to Table 1 (if it's editable) and placed below main and EV figure legends. Alternatively it can be uploaded as Table EV1 or included in the Appendix PDF as Appendix Table S1. The callout should be updated as appropriate callout; in either of the latter two cases, the table legend should be above the table.

Done.

15) Please ensure that all funding sources are entered into the manuscript submission system. Currently the following are missing in our submission system: the Max Planck Society; European Research Council (Starter Grant 3D-REG 101115401); via SFB 1565 (project 469281184/P02); the PhD program "Genome Science" - International Max Planck Research School at the Georg August University Göttingen; the

MSc/PhD program "Molecular Biology" - International Max Planck Research School at the Georg August University Göttingen

Done.

16) Synopsis:

- Synopsis image: Please provide a graphic that summarises the main findings of the manuscript on a glance and upload it as a high-resolution jpeg file 550 pixels wide x (300-600) pixels high.

Done.

- Synopsis text: Please provide a separate word document including a short standfirst (maximum of 300 characters, including spaces) and up to 5 bullet points to summarise the key NEW findings. They should be designed to be complementary to the abstract - i.e. not repeat the same text. We encourage inclusion of key acronyms and quantitative information (maximum of 30 words / bullet point). Please use the passive voice.

Done.

Done.

17) Source Data: Please ensure that a completed Source Data checklist is uploaded as a Related Manuscript File Instructions for Source Data and the Source Data checklist will be sent to you in a separate email shortly. Source Data should be organized as a single source data file (zipped) per figure for main figures (all EV and/or Appendix figure Source Data can be included in a single folder), with the panels clearly visible in the folder structure instead of a single excel file for all Source Data. e.g. all the Source data files for figure 1 need to be saved in a single folder and this needs to be zipped and then uploaded as "SD figure 1.zip" file.

Done. Please note that all main and EV figures are supported by deposited large scale data.

18) As part of the EMBO Publications transparent editorial process initiative (see our policy here: https://www.embopress.org/transparent-process#Review_Process), Molecular Systems Biology will publish online a Peer Review File (PRF) to accompany accepted manuscripts. This file will be published in conjunction with your paper and will include the anonymous referee reports, your point-by-point response and all pertinent correspondence relating to the manuscript. Let us know whether you agree with the publication of the PRF and as here, if you want to remove or not any figures from it prior to publication. Please note that the Authors checklist will be published at the end of the PRF.

We are happy with this arrangement.

19) After your paper is published, we may promote it on social media. If you have any handles or hashtags for Bluesky you would like included, please let us know.

[@clemauksch.bsky.social](https://bsky.app/profile/clemauksch.bsky.social)

@mariekeoudelaar.bsky.social

@mpi-nat.bsky.social

20) Please provide a point-by-point letter INCLUDING my comments and your detailed responses (as Word file).

Done.

Reviewer #1

Based on the revised manuscript and the accompanying point-by-point response, I find that the authors have addressed the reviewers' concerns in a generally thorough and satisfactory manner. They have provided direct experimental evidence with appropriate statistical support, which effectively clarifies the previously ambiguous claims regarding TAD insulation and nucleosome organization. Furthermore, the addition of a transcriptional quartile-based analysis lends a more structured rationale to the observed correlation between nucleosome perturbation and changes in chromatin architecture. Although the conclusions remain correlative rather than mechanistically definitive, the study offers conceptual novelty by delineating the role of nucleosome organization independently of CTCF/cohesin binding. Overall, the revisions have substantially improved the manuscript, bringing it to a standard appropriate for publication in Molecular Systems Biology.

We thank the Reviewer for their positive assessment of our revision.

Reviewer #2

The authors have provided a point-by-point response to reviewers that answers and clarifies most of their concerns.

The main criticisms of reviewers included:

1) If the small effects in genome topology (TAD boundary insulation and loop formation) detected upon FACT degradation were statistically significant and reproducible. The authors have added statistics, additional analysis grouping transcribed regions into different quartiles and additional replicates to confirm their findings.

2) If degradation of FACT did or did not disrupt CTCF and Cohesin binding from chromatin. The authors have revisited their CHIPmentation data set and analysis and confirmed that this is not the case.

3) A confusion regarding nucleosome positioning versus occupancy and if indeed the effects on these parameters were significant upon FACT degradation. The authors have corrected the terms and verified that indeed positioning is not altered but occupancy is, even if the reduction is small. The authors changed the title to "nucleosome organization", I would suggest calling it occupancy as this is the more accurate term as nucleosome organization can mean many different things.

We thank the Reviewer for their positive assessment of our revision. Regarding terminology, we would prefer to keep the term "nucleosome organization" in the title as an encompassing term for the significant effects we see on both nucleosome occupancy and fuzziness. However, we have included the more specific terms "nucleosome occupancy" and "nucleosome fuzziness" in the synopsis, so that the specific defects in nucleosome organization can immediately be appreciated by the readers.

6th Oct 2025

Manuscript number: MSB-2025-13164R

Title: FACT depletion demonstrates a role for nucleosome organization in TAD formation

Dear Dr. Oudelaar,

Congratulations on an excellent manuscript, I am pleased to inform you that your manuscript has been accepted for publication in Molecular Systems Biology. It has been a pleasure to work with you to get this to the acceptance stage.

Yours sincerely,

Sincerely,

Poonam Bheda, PhD
Scientific Editor
Molecular Systems Biology
